# LARGE-SCALE SPECTRAL GRAPH NEURAL NETWORKS VIA LAPLACIAN SPARSIFICATION

## ABSTRACT

Graph Neural Networks (GNNs) play a pivotal role in graph-based tasks for their proficiency in representation learning. Among the various GNN methods, spectral GNNs employing polynomial filters have shown promising performance on both homophilous and heterophilous graph structures. The scalability of spectral GNNs is limited because forward propagation requires multiple graph propagation executions, corresponding to the degree of the polynomial. On the other hand, scalable spectral GNNs detach the graph propagation and linear layers, allowing the message-passing phase to be pre-computed and ensuring effective scalability on large graphs. However, this pre-computation can disrupt end-to-end training, possibly impacting performance, and becomes impractical when dealing with high-dimensional input features. In response to these challenges, we propose a novel graph spectral sparsification method to approximate the propagation pattern of spectral GNNs. We prove that our proposed methods generate Laplacian sparsifiers for the random-walk matrix polynomial, incorporating both static and learnable polynomial coefficients. By considering multi-hop neighbor interactions into one-hop operations, our approach facilitates the use of scalable techniques. To empirically validate the effectiveness of our methods, we conduct an extensive experimental analysis on datasets spanning various graph scales and properties. The results show that our method yields superior results in comparison with the corresponding approximated base models.

## 1 INTRODUCTION

Graph Neural Networks (GNNs) have gathered increasing research attention because of their versatility in handling graph-structured data. They have demonstrated prominent performance in several kinds of real-world graph learning tasks, including link prediction (Sankar et al., 2021), recommendation systems (Wu et al., 2019b; Ying et al., 2018; Zhang et al., 2023), social analysis (Lin et al., 2022), drug discovery (Jiang et al., 2021), and traffic forecasting (Cui et al., 2020).

Spectral GNNs represent one fundamental branch of GNNs that works by constructing a graph filter in the spectral domain of the graph Laplacian matrix. This filtering mechanism enables the recombination of graph signals at different frequencies, effectively leveraging their spectral properties. Constrained by the impractical overhead of eigendecomposition, various works adopt distinct polynomial bases to approximate the desired filter operation, such as GPR-GNN (Chien et al., 2021) leverages a monomial basis, and BernNet (He et al., 2021) employs a Bernstein polynomial basis.

In general, spectral GNNs employing polynomial filters can be formally expressed as $\mathbf{Y} = g_{\mathbf{w}}(\mathbf{L}, f_\theta(\mathbf{X}))$, where $g_{\mathbf{w}}(\cdot)$ denotes the polynomial graph filter with coefficients $\mathbf{w}$, $f_\theta(\cdot)$ represents the linear layer with learnable parameters $\theta$, $\mathbf{L}$ is the Laplacian matrix, $\mathbf{X}$ and $\mathbf{Y}$ refer to the original node representation matrix and the output, respectively. Unlike GNNs that design a uniform aggregation function, spectral GNNs use polynomial filters to combine representations from $K$-hop neighbors, where $K$ is the polynomial degree. This property enables spectral GNNs to capture a broader scope of graphs and alleviates the dependence on "homophily assumption".

**Motivation.** Current spectral GNNs usually involve the computation of $\mathbf{L}^k f_\theta(\mathbf{X})$, which demands the full graph propagation during the $k$th forward propagation of training. However, existing acceleration devices, such as GPUs, suffer bottlenecks in storing the computational trees and node representation for large graphs. Drawing inspirations from SGC (Wu et al., 2019a), many works

such as ChebNetII (He et al., 2022) and OptBasis (Guo & Wei, 2023) detach the graph propagation phase from the linear layers to allow for the precomputation of $\mathbf{L}^k\mathbf{X}$. This transformation converts model training into the linear combination of $\mathbf{L}^k\mathbf{X}$ using learnable coefficient $w_k$ and linear layers. This process is independent of the graph structure and can be mini-batched naturally. Nevertheless, this method introduces the following defects with scalability: 1) Some researches (Lutzeyer et al., 2022; Busch et al., 2020) argue that separating training from the graph structure simplifies the network architecture and will negatively impact performance. 2) The training is no longer end-to-end due to the graph propagation preprocess. Emerging approaches (Chen et al., 2023; Chien et al., 2022) based on language models enhance the performance on Ogbn-papers100M, which requires the raw text as input and end-to-end training. 3) Preprocess is potentially impractical when dealing with large graphs with high-dimensional node features since the dimension of original features cannot be reduced by MLP. Given these considerations, it prompts the question: *Is there an approach to enhance the **scalability** of spectral GNNs **without decoupling** the graph propagation phase?*

**Contribution.** Inspired by Laplacian sparsification, we propose a novel approach to approximate the equivalent propagation matrix of filters in spectral methods. Specifically, we approximate $\widetilde{\mathbf{L}}_K \approx \sum_{k=0}^{K} w_k \mathbf{L}^k$, while keeping the number of non-zeros within an acceptable range. This sparsification effectively connects multi-hop neighbors by compressing the multi-step graph propagation, which enables the application of numerous scalable GNN algorithms (Hamilton et al., 2017; Zou et al., 2019; Fey et al., 2021). Importantly, it retains the integration of graph propagation within the model. Our contributions are summarized as follows:

- **Scalable method designing.** We propose a method termed *Graph Neural Networks with Laplacian Sparsification* (GNN-LS), which is the first work tackling the scalability issue of spectral GNNs to the best of our knowledge. This method offers various adaptions for different scenarios, including models with static polynomial coefficients, those with learnable polynomial coefficients, and a node-wise sampling way for semi-supervised tasks.

- **Theoretical analysis.** We provide rigorous mathematical proofs demonstrating that all the variations of our method construct Laplacian sparsifiers of $\sum_{k=0}^{K} w_k \mathbf{L}^k$ with a high probability and a low approximation error $\varepsilon$. This property ensures the quality and reliability of generated sparsifiers, guaranteeing the performance and efficiency of our model.

- **Extensive experiments.** We conduct comprehensive experiments aimed at validating the effectiveness and scalability of our methods, entangling with some spectral methods. The results consistently highlight the practical performance of our method, showcasing stable improvements compared to the corresponding baselines across diverse real-world datasets.

## 2 PRELIMINARIES

**Notations.** In this study, we consider the undirected graph $G = (V, E)$, where $V$ represents the node set and $E$ is the edge set. Let $n = |V|$ and $m = |E|$ denote the number of nodes and edges, respectively. $\mathbf{A} \in \{0, 1\}^{n \times n}$ represents the adjacency matrix of the graph $G$. The diagonal degree matrix is denoted by $\mathbf{D}$, and $\mathbf{D}_{ii} = \sum_j \mathbf{A}_{ij}$. The normalized adjacency matrix and normalized graph Laplacian of $G$ are defined as $\mathbf{P} = \mathbf{D}^{-1/2}\mathbf{A}\mathbf{D}^{-1/2}$ and $\mathbf{L} = \mathbf{I}_n - \mathbf{D}^{-1/2}\mathbf{A}\mathbf{D}^{-1/2}$, respectively. Note that the normalized graph Laplacian $\mathbf{L}$ is symmetric and positive semidefinite. We express the eigenvalue decomposition of $\mathbf{L}$ as $\mathbf{U}\boldsymbol{\Lambda}\mathbf{U}^\top$, where $\mathbf{U}$ is a unitary matrix containing the eigenvectors, and $\boldsymbol{\Lambda} = \text{diag}\{\lambda_1, \lambda_2, \ldots, \lambda_n\}$ comprises the eigenvalues of $\mathbf{L}$. We usually modify the Laplacian with $\widehat{\mathbf{L}} = \frac{2\mathbf{L}}{\lambda_{\max}} - \mathbf{I}$ to scale the eigenvalues to $[-1, 1]$. Note that the $\lambda_{\max}$ is set to 2 in practice, then we have $\widehat{\mathbf{L}} \approx -\mathbf{P}$.

**Spectral GNNs.** Spectral-based GNNs exploit the spectral attributes of $G$ and apply the graph convolution operation on the spectral domain. Many works (Klicpera et al., 2019; He et al., 2022) either approximate the filter with polynomial or exhibit similar properties of polynomial filters. The graph filtering operation with respect to the graph Laplacian matrix $\mathbf{L}$ and signal $\mathbf{x}$ is defined as

$$h(\mathbf{L}) * \mathbf{x} = \mathbf{U}h(\boldsymbol{\Lambda})\mathbf{U}^\top \mathbf{x} \approx \mathbf{U}\left(\sum_{k=0}^{K} w_k \boldsymbol{\Lambda}^k\right)\mathbf{U}^\top \mathbf{x} = \left(\sum_{k=0}^{K} w_k \mathbf{L}^k\right)\mathbf{x}, \tag{1}$$

where $\mathbf{w} = [w_0, w_1, \ldots, w_k]$ represents the polynomial coefficient vector.

**Homophily.** Homophily measures the tendency of the connected nodes on the graph to have the same label in node classification tasks. This property heavily influences the classic GNN models which utilize one-hop neighbors, while the spectral GNNs can leverage multi-hop neighbor importance. We usually quantify the homophily of a graph with the following edge homophily:

$$\mathcal{H}(G) = \frac{1}{m} \left| \{(u,v) : y(u) = y(v) \wedge (u,v) \in E\} \right|,$$

where $y(\cdot)$ returns the label of nodes. Intuitively, $\mathcal{H}(\cdot)$ denotes the ratio of homophilous edges on the graph. A heterophilous graph implies $\mathcal{H}(G) \rightarrow 0$.

## 3 PROPOSED METHOD

### 3.1 MOTIVATION

If we take retrospect on the spectral GNNs, this category of GNNs yields promising results, especially when applied to heterophilous datasets. However, an aspect that has received less attention in existing spectral GNN research is scalability. Numerous works(Hamilton et al., 2017; Zeng et al., 2020) that prioritize scalability resort to random sampling techniques to reduce the neighborhood of central nodes, or employ methods like historical embedding (Fey et al., 2021) to approximate node embeddings. However, the convolution of spectral GNNs gathers information from up to $K$-hop neighbors, posing challenges to the effective deployment of scalable techniques.

Nevertheless, it is still possible to enhance the scalability of spectral GNNs using straightforward methods. Following SGC (Wu et al., 2019a), many of the spectral methods (He et al., 2022; Guo & Wei, 2023) decouple the graph propagation from training to simplify the network architecture. During the graph propagation, they compute $\mathbf{L}^k \mathbf{x}$ as described in Equation 1, which can be preprocessed on the CPU. Consequently, the learnable parameters control the linear combination and transformation of the propagated embeddings. Since the graph structure is only used in graph propagation, the training naturally lends itself to mini-batching.

However, this method is not without its drawbacks. First, the decoupling of the GNNs leads to a non-end-to-end model. We cannot use the learnable language models to enhance the node representations with the raw text of large-scale datasets like Ogbn-papers100M. Second, we cannot apply a learnable linear layer to reduce the dimension of raw features. The incoming dataset must be preprocessed with additional CPU time and storage space. Third, the performance of SGC has been proven to suffer a loss in comparison with GCN. Given these considerations, the question arises: Is it feasible to enhance the scalability of spectral GNNs without decoupling the network architecture?

### 3.2 SIMPLIFY THE GRAPH PROPAGATION WITH LAPLACIAN SPARSIFICATION

When we revisit Equation 1, we observe that the filter step can be reformulated as a matrix multiplication involving the combined powers of Laplacian (referred to as $\mathbf{L}_K = \sum_{k=0}^{K} w_k \mathbf{L}^k$, Laplacian polynomial) and the input signal. If we obtain the Laplacian polynomial $\mathbf{L}_K$, the propagation is then squeezed to a single step instead of multi-hop message-passing. Since the computation and storage of matrix $\mathbf{L}_k$ is overwhelming, we introduce the Laplacian sparsification (Spielman & Teng, 2011) to approximate the matrix within high probability and tolerable error.

Laplacian sparsification is designed to create a sparse graph that retains the spectral properties of the original graph. In essence, the constructed graph, with fewer edges, is spectrally similar to the original one. We provide a formal definition of the spectrally similar as follows.

**Definition 3.1.** *Given an weighted, undirected graph $G = (V, E, w)$ and its Laplacian $\mathbf{L}_G$. We say graph $G'$ is spectrally similar to $G$ with approximation error $\varepsilon$ if we have*

$$(1 - \varepsilon)\mathbf{L}_{G'} \preccurlyeq \mathbf{L}_G \preccurlyeq (1 + \varepsilon)\mathbf{L}_{G'},$$

*where we declare that two matrix $\mathbf{X}$ and $\mathbf{Y}$ satisfy $\mathbf{X} \preccurlyeq \mathbf{Y}$ if $\mathbf{Y} - \mathbf{X}$ is semi-definite.*

A well-established result states that $\mathbf{X} \preccurlyeq \mathbf{Y}$ implies $\lambda_i(\mathbf{X}) \leq \lambda_i(\mathbf{Y})$ for each $1 \leq i \leq n$, where $\lambda_i(\cdot)$ denotes the $i$th largest eigenvalue of the matrix. This corollary reveals that when two matrices,

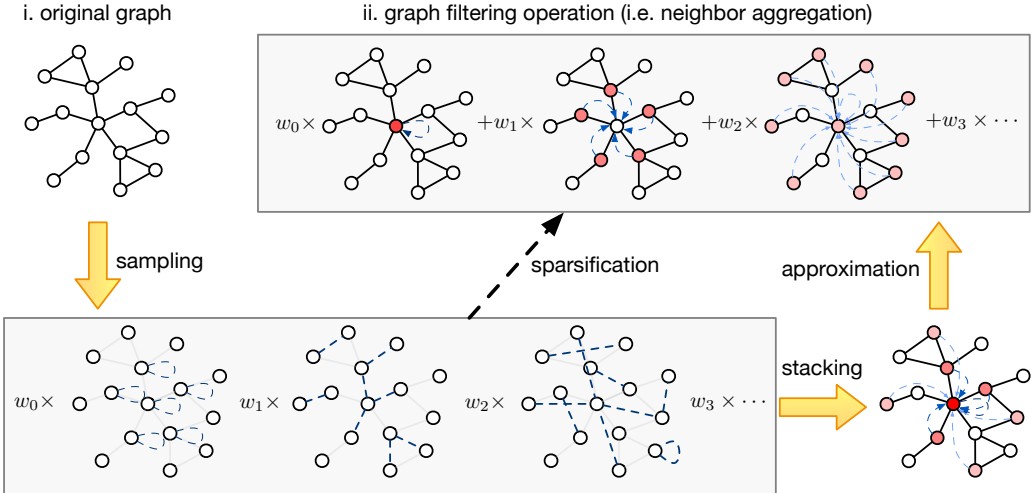

Figure 1: An overview of how Laplacian sparsification works. For clarity, the propagation of one single center node is illustrated. Laplacian sparsification is applied to the entire graph, generating fully sparsified graphs to satisfy the propagation requirements of all the nodes in the graph.

---

**Algorithm 1:** Edge Sampling of $\mathbf{D}(\mathbf{D}^{-1}\mathbf{A})^k$

---

**Input:** Edge set $E$, power index $k$.
**Output:** Sampled edge $(u, v)$
1 $e \leftarrow$ sample an edge from $E$ uniformly at random
2 $i \leftarrow$ sample an integer in $[0, k-1]$ uniformly at random
3 $u \leftarrow$ the end of random walk on $E$ (i.e. graph $G$), starting from $e_u$ with length $i$
4 $v \leftarrow$ the end of random walk on $E$ (i.e. graph $G$), starting from $e_v$ with length $k - i - 1$
5 **return** $(u, v)$

---

such as $\mathbf{L}_G$ and $\mathbf{L}_{G'}$, exhibit spectral similarity, their quadratic form and eigenvalues are in close correspondence Graph sparsification is a fundamental problem of graph theory. Many studies (Batson et al., 2013) have introduced algorithms capable of generating $\varepsilon$-sparsifiers for a given graph.

The sole difference between scaled Laplacian and the propagation matrix is a negative sign, we have

$$\mathbf{L}_K = \sum_{k=0}^{K} w_k \mathbf{L}^K \approx \sum_{k=0}^{K} w_k \mathbf{P}^k = \mathbf{D}^{-1/2} \cdot \mathbf{D}\left(\sum_{k=0}^{K} w_k \left(\mathbf{D}^{-1}\mathbf{A}\right)^k\right)\mathbf{D}^{-1/2}, \qquad (2)$$

where the negative sign can incorporated into the coefficients $w_k$. From Equation 2, we observe that we may convert our desiring matrix $\mathbf{L}_K$ to a random walk matrix polynomial $\left(\mathbf{D}^{-1}\mathbf{A}\right)^k$ with coefficients $[w_0, ..., w_K]$. Following the prior works (Spielman & Srivastava, 2008; Cheng et al., 2015), we have a promising guarantee to approximate random walking matrix polynomial by effective resistance termed Theorem 3.2.

**Theorem 3.2.** *(Random walk polynomial sparsification.)* *For any weighted, undirected graph $G$ with $n$ vertices and $m$ edges, any $\mathbf{w} = [w_0, w_2, ..., w_K] \in \mathcal{R}^{(K+1)}, \mathbf{w} \neq \mathbf{0}$ and any approximation parameter $\varepsilon$, we can construct an $\varepsilon$-sparsifier of the random walk matrix polynomial $\sum_{k=0}^{K} w_k \mathbf{D}\left(\mathbf{D}^{-1}\mathbf{A}\right)^k$ within $O(n \log n/\varepsilon^2)$ edges and probability $1 - K/n$ at least.*

We have extended the original theorem proposed by (Cheng et al., 2015) to accommodate non-normalized polynomial coefficients $\mathbf{w}$. Sampling an edge from $\mathbf{D}(\mathbf{D}^{-1}\mathbf{A})^k$ is an atomic operation in constructing our desired graph, and it is frequently employed in our subsequent algorithms. We state this procedure in Algorithm 1. In the upcoming sections, we will present our comprehensive algorithms, which include the weight correction and the adaptation for both static and learnable polynomial coefficients $\mathbf{w}$. For in-depth discussion regarding the approximation estimation, correctness proof, and complexity analysis, please retrieve section 4 and appendix A.1.

---

**Algorithm 2:** Static Laplacian Sparsified Graph Construction (SLSGC)

---

**Input:** Hop weights $\mathbf{w}$.
**Input: Model Saved:** Vertice set $V$, edge set $E$, degrees $\mathbf{d}$, maximum neighbor hop $K$, total
     sampling number $M$.
**Output:** Weighted edge set $\widetilde{E}$ after laplacian sparsification.

**1** $m \leftarrow |E|$
**2** $\widetilde{E} \leftarrow \emptyset$
**3 for** $i$ from 1 to $M$ **do**
**4**     $k \leftarrow$ sample an integer $k$ from distribution $\Pr\{k = j\} = |w_j|/\|\mathbf{w}\|_1$
**5**     $(u, v) \leftarrow$ Edge_Sampling$(E, k)$
**6**     $\widetilde{E} \leftarrow \widetilde{E} \cup \left( (u, v), \text{sgn}(w_k)\|\mathbf{w}\|_1 d_u^{-1/2} d_v^{-1/2} \cdot \frac{m}{M} \right)$

**7 return** $\widetilde{E}$

---

### 3.2.1 LAPLACIAN SPARSIFICATION FOR STATIC POLYNOMIAL COEFFICIENTS

Some of the early classic GNNs, like GCN (Kipf & Welling, 2017) and APPNP (Klicpera et al., 2019), employ static Laplacian polynomials as the graph filter. For example, the propagation layer of APPNP fixes the weight $w_k = \alpha(1 - \alpha)^k, k \neq K$, and $w_K = (1 - \alpha)^K$ for different hops.

Note that if we repeat Algorithm 1 for $M$ times with the same power index $k$ and apply each obtained edge with the weight $m/M$, we can approximate $\mathbf{D}(\mathbf{D}^{-1}\mathbf{A})^k$. This result is still distant from our desired form in Equation 2. First, we need to approximate the random walk matrix polynomial based on our existing method of approximating a term $\mathbf{D}(\mathbf{D}^{-1}\mathbf{A})^k$. One intuitive and efficient idea is to distribute the $M$ edges among all $K + 1$ subgraphs of $\mathbf{D}(\mathbf{D}^{-1}\mathbf{A})^k$. The number of edges assigned to each subgraph follows a multinomial distribution with weights $\mathbf{w}$. Note that $w_k$ is not guaranteed to be positive. The absolute value of $w_k$ is proportional to the probability of being sampled, while $\text{sgn}(w_k)$ decides the sign of the edge weight. We select a random walk length $k$ based on the probability distribution $\Pr\{k = i\} = |w_i|/\|\mathbf{w}\|_1$. Second, we execute Algorithm 1 with edge set $E$ and power index $k$ to generate an edge $(u, v)$. Given that we are now approximating the graph $\mathbf{D}(\mathbf{D}^{-1}\mathbf{A})^k$, the edge value is adjusted to $\text{sgn}(w_k)\|\mathbf{w}\|_1 \cdot d_u^{-1/2} d_v^{-1/2} \cdot m/M$.

We provide the pseudo-code for constructing a Laplacian sparsified random walk matrix polynomial with static coefficients in Algorithm 2. Additionally, we offer an example of such a model integrated with our method APPNP-LS in Appendix A.2.1.

### 3.2.2 LAPLACIAN SPARSIFICATION FOR LEARNABLE POLYNOMIAL COEFFICIENTS

Several recent spectral works, like GPR-GNN (Chien et al., 2021), BernNet (Wu et al., 2019b), and ChebNetII (He et al., 2022), employ learnable polynomial coefficients to dynamically adapt the proper filter. For example, GPR-GNN uses monomial bases to identify the optimal filter, while BernNet employs Bernstein polynomial basis to align with the property that the eigenvalues of normalized Laplacian fall within the range $[0, 2]$.

Upon revisiting Algorithm 2, it becomes apparent that the polynomial coefficients $\mathbf{w}$ primarily affect the sampling of the power index $k$ and the adjustment of edge weights. To facilitate the training of $\mathbf{w}$, we need to calculate the derivative of each $w_k$ for gradient descent. However, following Algorithm 2, we cannot obtain the correct derivatives of $\mathbf{w}$ since all the $w_k$ equally contribute to the part that generates gradients. To address this issue, we directly multiply the polynomial coefficients $\mathbf{w}$ with the weight of the sampled edges instead of the sampling with probability $|w_k|/\|\mathbf{w}\|_1$ in SLSGC. Thus, the edge weight becomes $w_k d_u^{-1/2} d_v^{-1/2} \cdot m/M$. This adjustment connects the gradient of $\mathbf{w}$ with the message passed by the corresponding edges, ensuring the correct derivative chain for training $\mathbf{w}$. However, this modification splits coefficients $w_k$, leading to independent sampling for each hop $k$ and potentially sacrificing the efficiency. Theoretically, we need to sample more edges to support the training of $\mathbf{w}$ while maintaining the bound of approximation.

We provide the pseudo-code of constructing a Laplacian sparsified random walk matrix polynomial with learnable coefficients (named GLSGC) in Appendix A.2.2. Besides, we offer an example of such a model integrated with our method GPR-LS in Appendix A.2.2

---

**Algorithm 3:** Edge Sampling by Effective Resistance

---

**Input:** Edge set $E$, upper bound of effective resistance $R_{\text{sup}}$, sampling number $M$.

**Output:** Sampled edge set $\widetilde{E}$

**1** $\widetilde{E} \leftarrow \emptyset$

**2 for** $i$ from 1 to $M$ **do**

**3** $\quad\big\lfloor\ \widetilde{E} \leftarrow \widetilde{E} \cup$ sampled edge $e$ with probability $p(e) \propto w(e)R_{\text{sup}}(e)$ and weight $1/(M \cdot R_{\text{sup}}(e))$

**4 return** $E$

---

### 3.3 NODE-WISE LAPLACIAN SAMPLING FOR SEMI-SUPERVISED TASKS

For representation learning tasks on large-scale graphs with few training nodes, classic S/GLSGC are wasteful as most edges are sampled between nodes in the validation/test set. To address this issue, we propose a node-wise sampling method to approximate the corresponding rows of the result of Equation 2 for the training nodes. This enhancement significantly improves the training efficiency, as nodes in the validation/test set will not aggregate information during training.

Reflecting on Equation 2, it becomes clear that $((\mathbf{D}^{-1}\mathbf{A})^k)_{i,j}$ determines the probability of a random walk of length $k$ starting from node $i$ and ending at node $j$. This inspires us that we can ensure at least on incident node of the sampled edge belongs to the training set by directly sampling the random walk from the training set. Following Equation 2, for each random walk length $k$, we first distribute the $M$ sampled edges among all nodes in proportion to their degrees. Then we perform random walk samplings and correct the value of each generated edge $(u_0, u_k)$ to $d_{u_0}^{-1/2} d_{v_0}^{-1/2} \cdot w_k / M$ for each sampled walk $(u_0, u_1, ..., u_k)$.

It can be mathematically proven that the proposed algorithm produces an unbiased approximation of the selected rows of $\mathbf{D}(\mathbf{D}^{-1}\mathbf{A})^k$. Moreover, this method is node-wise, allowing for natural mini-batching. As a result, we achieve a strong scalability promotion of the spectral methods. The pesudo-code of this algorithm is proposed in Appendix 8

## 4 THEORETICAL ANALYSIS

In this section, we will conduct a comprehensive analysis, including rigorous proofs of correctness, complexity, and other key properties of the methods we have introduced. Our mathematical analyses closely follow the theoretical foundations proposed by Cheng et al. (2015).

From a graph theory perspective, a weighted graph has a close relationship with electric flow. Each edge with weight $w(e), e = (u, v) \in E$ ($w(e) = 1$ for an unweighted graph) can be considered equivalent to a resistor with resistance $1/w(e)$ connecting nodes $u$ and $v$. When we view the graph as a large complex resistor network, the resistance between $u$ and $v$ is defined as the effective resistance $R(u, v)$.

**Theorem 4.1.** *(Cheng et al., 2015) Given a weighted graph $G$ and the upper bound of its effective resistance $R_{sup}(e) \geq R(e)$. For any approximation parameter $\varepsilon$, there exists a sampled graph $\widetilde{G}$ with $M = O\left(\log n/\varepsilon^2 \cdot \left(\sum_{e \in E} w(e)R_{sup}(e)\right)\right)$, satisfying $(1 - \varepsilon)\mathbf{L}_G \preccurlyeq \mathbf{L}_{\widetilde{G}} \preccurlyeq (1 + \varepsilon)\mathbf{L}_G$ with at least $1 - 1/n$ probability.*

This theorem marks the initial step, whose pseudo-code of the algorithm is presented at Algorithm 3. Even though our focus is on an unweighted graph $G$, the final destination of our approximation $\mathbf{D}(\mathbf{D}^{-1}\mathbf{A})$ is inherently weighted. The weight between node $u$ and $v$ on the graph $G_r$ with its adjacency matrix $\mathbf{A}_r = \mathbf{D}(\mathbf{D}^{-1}\mathbf{A})^r$ can be considered as the union of all the paths between nodes $u$ and $v$ with length $r$ showing up in the unweighted graph $G$, i.e. $\mathbf{A}_r(u, v) = \sum_{p \in P} w(p)$, where $P = \{(u = i_0, i_1, \cdots, i_r - 1, v = i_r) | (i_j, i_{j+1}) \in E, j = 0, 1, \cdots, r - 1\}$.

**Lemma 4.2.** *(Cheng et al., 2015) **(Upper bound of effective resistance.)** The effective resistance between two nodes $u$ and $v$ on graph $G_r$ is upper bounded by*

$$R_{G_r}(u, v) \leq \sum_{j=0}^{r-1} \frac{2}{\mathbf{A}(i_{j-1}, i_j)},$$

*where $(u = i_0, i_1, \cdots, i_{r-1}, v = i_r)$ is a path on $G$.*

This is a general version of the upper bound of effective resistance about the graph $G_r$. As we are primarily concerned with unweighted graph $G$, $R_{\sup,G_r}(u,v)$ can be simplified to a constant $2r$. We can prove a more robust conclusion that Algorithm 1 draws path $p$ on graph $G$ with the probability **strictly** proportional to $w(p)$. This probability is independent of the $R_{\sup,G_r}(\cdot)$, which means that running a Monte-Carlo sampling on the graph yields an **unbiased approximation** of $G_r$.

Therefore, by replacing the sampling process in Algorithm 3 with Algorithm 1, we obtain an unbiased graph sparsifier of $G_r$ with $O(rm \log n/\varepsilon^2)$ edge. This sparsifier can be further reduced to $O(n \log n/\varepsilon^2)$ by the existing works []. Both our proposed methods, SLSGC and GLSGC, follow the complexity outlined in Theorem 3.2. The detailed proof is proposed in Appendix A.1. Note that the proposed bound is much tighter than what is practically required. In practice, the sampling number can be set rather small to achieve the desired performance.

In the method proposed in Section 3.3, the sampled random walk of length $r$ originating from a distinct source node $u$ also shares the same upper bound of effective resistance $2r$, which can be considered as the effective resistance based Laplacian sparsification. However, our method is more intuitive and simplified since we directly sample the desired edge in proportional to $w(p)$, without relying on effective resistance. We take consideration of the single candidate start $u$ for the random walks and define $c_r(u,v)$ as the final generated edge weight of $(u,v)$ on graph $G_r$.

**Theorem 4.3.** *Given a weighted graph G, and the candidate set U of the random walk starts. For any $u \in U$ and $v \in V$, we have $c_r(u,v)$ is the unbiased approximation of $\left(\mathbf{D}(\mathbf{D}^{-1}\mathbf{A})^r\right)_{u,v}$.*

The detailed proof is presented in Appendix A.1. Similarly, we do not need a large number of samples to achieve superior performance in practice. This node-centered sampling method enables us to adapt our method to semi-supervised tasks, saving the memory for a larger batch size.

## 5 RELATED WORKS

**Spectral GNNs.** To align with the proposed methods above, we categorize the spectral GNNs based on their used polynomial filter: static (predefined) polynomial filters and learnable polynomial filters. For static polynomial filters, GCN (Kipf & Welling, 2017) uses a fixed simplified Chebyshev polynomial approximation and operates as a low-pass filter. APPNP (Klicpera et al., 2019) combines the GCN with Personalized PageRank, which can approximate more types of filters but still cannot operate as an arbitrary filter. GNN-LF/HF (Zhu et al., 2021) predefines the graph filter from the perspective of graph optimization. For learnable polynomial filters, ChebNet (Defferrard et al., 2016) first approximates the desired filter with the Chebyshev polynomial base, which can operate as an arbitrary filter theoretically. Similarly, GPRGNN (Chien et al., 2021) considers the monomial base to learn the importance of $k-$hop neighbors directly. BernNet (He et al., 2021) utilizes the Bernstein polynomial base to make the filter semi-positive definite. ChebNetII (He et al., 2022) revisits the ChebNet and proposes a new filter design via Chebyshev interpolation. Guo & Wei (2023) proposes FavardGNN to learn a basis from all possible orthonormal bases and OptBasisGNN to compute the best basis for the given graph.

**Graph Sparsification.** Graph sparsification is a consistently studied topic in graph theory. Spielman & Teng (2004) introduces the concept of graph sparsification and presents an efficient algorithm to approximate the given graph Laplacian with a smaller subset of edges while maintaining the spectral properties. Spielman & Srivastava (2008) leverages effective resistance to yield the promising random sampled edges on the graph. Lee & Sun (2015) proposes the first method to construct linear-sized spectral sparsification within almost linear time. Our work is mainly enlightened by the works (Qiu et al., 2019) and (Cheng et al., 2015), which make an approximation to the series of the multi-hop random walk sampling matrix.

## 6 EXPERIMENTS

### 6.1 TESTED MODELS, DATASETS, AND CONFIGURATIONS

**Tested models.** We compare our method with the vanilla MLP, classic GNNs like GCN, GC-NII (Chen et al., 2020), and GAT (Velickovic et al., 2018), detaching methods like SGC, spectral

Table 1: Experimental results of some baselines and our Laplacian sparsification entangled methods on multiple **small-scale datasets**. Model name with suffix "-LS" represents the spectral methods entangled with our proposed method. The lines beginning with "Δ" reveal the performance difference between the base model and its variation. Most evaluation metrics are accuracy(%), but ROC AUC (%) for datasets with 2 classes (Twitch-de, and Twitch-gamers / Penn94 in Table 2). The **bold** font highlights the best results, whereas the underlined numbers indicate the second and third best.

| Dataset | Cora | Cite. | PubM. | Actor | Wisc. | Corn. | Texas | Photo | Comp. | T-de |
|---|---|---|---|---|---|---|---|---|---|---|
| $\mathcal{H}(G)$ | 0.810 | 0.736 | 0.802 | 0.219 | 0.196 | 0.305 | 0.108 | 0.827 | 0.777 | 0.632 |
| MLP | 76.72 | 77.29 | 86.48 | 39.99 | 90.75 | 92.13 | 92.11 | 90.11 | 85.00 | 68.84 |
| GAT | 86.80 | 81.15 | 86.61 | 35.26 | 69.13 | 71.97 | 79.02 | 93.31 | 88.39 | 67.90 |
| GCNII | 88.52 | 81.24 | 89.17 | 41.20 | 82.88 | 90.49 | 84.75 | 94.20 | 88.55 | 68.03 |
| GCN | 87.78 | 81.50 | 87.39 | 35.62 | 65.75 | 71.96 | 77.38 | 93.62 | 88.98 | 73.72 |
| SGC | 87.24 | 81.53 | 87.17 | 34.40 | 67.38 | 70.82 | 79.84 | 93.41 | 88.61 | 73.70 |
| Δ | -0.54 | +0.03 | -0.22 | -1.22 | +1.63 | -1.14 | +2.46 | -0.21 | -0.37 | -0.02 |
| GPR | 88.80 | 81.57 | **90.98** | 40.55 | 91.88 | 89.84 | **92.78** | 95.10 | 89.69 | **73.91** |
| GPR-LS | **89.31** | 81.65 | 90.95 | 41.82 | **93.63** | 91.14 | 92.62 | **95.30** | **90.47** | 73.49 |
| Δ | +0.51 | +0.08 | -0.03 | +1.27 | +1.75 | +1.30 | -0.16 | +0.20 | +0.78 | -0.42 |
| APPNP | 88.69 | 81.32 | 88.49 | 40.73 | 90.38 | 90.98 | 90.82 | 93.82 | 86.97 | 68.28 |
| APPNP-LS | 88.44 | **82.28** | 88.70 | **41.98** | 91.00 | **92.30** | 90.98 | 93.79 | 87.84 | 72.82 |
| Δ | -0.25 | +0.96 | +0.21 | +1.25 | +0.62 | +1.32 | +0.16 | -0.03 | +0.87 | +4.54 |

GNN with static polynomial coefficients like APPNP, and spectral GNN with learnable coefficients GPRGNN. For the APPNP and GPR-GNN, we entangle them with our proposed Laplacian sparsification method for the effectiveness test. All the baselines are reimplemented with Pytorch (Paszke et al., 2019) and PyG (Fey & Lenssen, 2019) library modules as competitors. Detailed parameter matrix and reproduction guidance are stated in Appendix A.4.

**Datasets.** All the baselines and our proposed method are tested with various datasets with diverse homophily and scales. Cora, Citeseer, and PubMed (Sen et al., 2008; Yang et al., 2016) have been the most widely tested citation networks since the emergence of the GCN. Photos and computers (McAuley et al., 2015; Shchur et al., 2018) are the segment of the co-purchase graphs from Amazon. Actor (Pei et al., 2020) is the co-occurrence graph of film actors on Wikipedia. Cornell, Texas, and Wisconsin (Pei et al., 2020) are the webpage hyperlink graphs from WebKB. Twitch-de and Twitch-gamers (Lim et al., 2021) are the social network of Twitch users. Penn94 (Lim et al., 2021) is a friendship network from the Facebook 100 networks. Ogbn-arxiv and Ogbn-papers100M (Hu et al., 2020) are two public datasets proposed by the OGB team. The downstream task of these datasets is node classification. Detailed information on the datasets is stated in Appendix A.3.

**Configurations.** All the experiments except those with dataset Ogbn-papers100M are conducted on the server equipping GPU NVIDIA A100 40GB. For Ogbn-papers100M, we deploy our experiment on the server with GPU NVIDIA A100 80GB and 512G RAM. Detailed information on experimental environments and package versions are stated in Appendix A.4.1.

## 6.2 RESULTS ON SMALL-SCALE REAL-WORLD DATASETS

In this section, we conduct full-supervised transductive node classification tasks on 10 small-scale real-world datasets. The detailed results are presented in the Table 1. Please note that we evaluate the relative performance change between the original base model and its variant, with a clear distinction marked by the horizontal line.

We first examine the comparison of GCN and SGC. SGC can be seen as the detached version of GCN, which first executes a $K$-hop graph propagation ($K = 2$ in practice), followed by the linear layers. Our empirical findings verify that the detaching manner does exert a negative influence on the performance of GCN. Despite SGC exhibiting occasional performance improvements on specific small-scale datasets, the overall results are still distant from the powerful baseline GPR.

The remaining entries in Table 1 offer an insightful comparison between some well-established GNN models, chosen spectral works, and their Laplacian sparsified variation. GPR-GNN (abbreviated as GPR) is one of the strongest spectral GNNs with a learnable polynomial filter, making our evaluation promising. The practical performance of GPR-GNN with Laplacian sparsification is superior to its original version, even under the potential risk of effect loss caused by the approximation.

Table 2: Experimental results of some baselines and our Laplacian sparsification entangled methods on multiple **medium- and large-scale datasets**. Proposed results shares the same annotation formats with Table 1, whereas asterisk(*) denotes that the result is generated by detached models.

| Dataset | Ogbn-arxiv | Twitch-gamers | Penn94 | Ogbn-papers100M |
|---|---|---|---|---|
| $\mathcal{H}(G)$ | 0.655 | 0.554 | 0.470 | - |
| MLP | 54.04 | 64.90 | 84.17 | 41.36 |
| GAT | 70.45 | 62.47 | 74.14 | (GPU OOM) |
| GCNII | 70.64 | 63.10 | 75.19 | (GPU OOM) |
| GCN | 70.18 | **67.86** | 82.55 | (GPU OOM) |
| SGC | 70.35 | 66.87 | (GPU OOM) | 59.26* |
| $\Delta$ | +0.17 | -0.99 | - | - |
| GPR | **71.03** | 66.69 | 82.92 | 61.68* |
| GPR-LS | 70.32 | 66.45 | **85.00** | 61.76 |
| $\Delta$ | -0.71 | -0.24 | +2.08 | +0.08 |
| APPNP | 69.87 | 65.11 | 82.89 | (GPU OOM) |
| APPNP-LS | 69.08 | 65.83 | 83.17 | 62.10 |
| $\Delta$ | -0.79 | +0.72 | +0.28 | - |

We choose APPNP as the typical baseline for those spectral works with static polynomial filters. In our implementation, the number of sampled edges for the Laplacian sparsification in spectral works with static polynomial filters is a mere fraction (specifically, $k$ times less) when compared to those with learnable filters. Surprisingly, APPNP-LS exhibits equivalent or even superior performance compared to the original APPNP while achieving a similar performance elevation to the GPR series, despite employing far fewer sampled edges.

### 6.3 RESULTS ON LARGE-SCALE REAL-WORLD DATASETS

To further assess the scalability of both our models and the baselines, we conduct experiments on a diverse range of medium- and large-scale real-world datasets. Note that the dataset Penn94 contains high-dimensional original node representation, and Ogbn-papers100M includes an extensive network with over $10^8$ nodes and $10^9$ edges. The experimental results are presented in Table 2.

As is shown in Table 2, our models consistently deliver competitive results even as we scale up to large datasets. Interestingly, on the heterophilous datasets proposed by Lim et al. (2021), our models show a slight performance advantage over the corresponding base models, despite the inherent approximation imprecision. This phenomenon shows that the Laplacian sparsification possesses the outstanding ability in 1) denoising, i.e. sampling the unnecessary neighbor connections with low probabilities, and 2) approximating the desired complex filters tailored to the heterophilous graphs.

As data scales up, many existing models suffer scalability challenges. For instance, the standard SGC cannot execute forward propagation on Penn94 without preprocessing since Penn94 contains 1.4 million edges and 4,814 dimensions of original features. The graph propagation leads to GPU out-of-memory errors without the dimension reduction. The dashed lines in Table 2 signify instances where the model cannot be trained on our devices. For SGC and GPRGNN, we preprocess the graph propagation of the required hops on Ogbn-papers100M. Our results clearly demonstrate that GPR-LS attains the performance of the corresponding decoupling method. APPNP-LS allows APPNP to function normally within limited storage space. These outcomes validate the effectiveness of our proposed method in significantly enhancing the scalability of conventional spectral approaches.

## 7 CONCLUSION

In this paper, we present a novel method centered on Laplacian sparsification to significantly enhance the scalability of spectral GNNs. Our approach demonstrates its capability to approximate the equivalent propagation matrix of Laplacian filters, making it compatible with existing scalable techniques. We provide the theoretical proof affirming that our model produces the correct sparsifier with a high probability, low approximation parameter $\varepsilon$, and an acceptable number of non-zeros in the propagation matrix. The experimental results validate that our methods yield comparable or even superior performance compared to the corresponding base models. This remarkable achievement is particularly noteworthy considering that we sample far fewer edges than the theoretical bound, underscoring the exceptional ability of our method to approximate desired filters.

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

## A APPENDIX

### A.1 PROOFS OF THE PROPOSED THEORIES

#### A.1.1 DETAILED ANALYSIS ABOUT SLSGC AND GLSGC ALGORITHM

In this section, we begin by restating the previously established conclusion. The weight of a path, denoted as $p = (u_0, u_1, ..., u_r)$ can be formally expressed as

$$w(p) = \frac{\prod_{i=0}^{r-1} \mathbf{A}_{u_{i-1}, u_i}}{\prod_{i=1}^{r-1} \mathbf{D}_{u_i, u_i}}.$$

This value is symmetrical when viewed from $u_0$ and $u_r$, which is a slight deviation from the random walk probability, as it encompasses the probability of the walk starting from the initial node.

The upper bound of effective resistance for a path $p$ on the weighted graph can be expressed as

$$R_{\sup, G_r}(u_0, u_r) = \sum_{i=0}^{r-1} \frac{2}{\mathbf{A}_{u_i, u_{i+1}}},$$

where $p$ is a path within graph $G$, startingfrom $u_0$, ending at $u_r$, and possessing a length of $r$.

As proposed by Cheng et al. (2015), we restate the conclusion that

$$\sum_{p=(u_0,\cdots,u_r)} w(p) R_{\sup,G_r}(u_0,u_r) = \sum_p \left( \sum_{i=0}^{r-1} \frac{2}{\mathbf{A}_{u_i,u_{i+1}}} \right) \left( \frac{\prod_{j=0}^{r-1} \mathbf{A}_{u_j,u_{j+1}}}{\prod_{j=1}^{r-1} \mathbf{D}_{u_j,u_j}} \right)$$

$$= 2 \sum_p \sum_{i=1}^r \left( \frac{\prod_{j=1}^{i-1} \mathbf{A}_{u_{j-1},u_j} \prod_{j=i}^{r-1} \mathbf{A}_{u_j,u_{j+1}}}{\prod_{j=1}^{r-1} \mathbf{D}_{u_j,u_j}} \right)$$

$$= 2 \sum_{e \in E} \sum_{i=1}^r \left( \sum_{p|(u_{i-1},u_i)=e} \frac{\prod_{j=1}^{i-1} \mathbf{A}_{u_{j-1},u_j}}{\prod_{i=1}^{j-1} \mathbf{D}_{u_j,u_j}} \cdot \frac{\prod_{j=i}^{r-1} \mathbf{A}_{u_j,u_{j+1}}}{\prod_{j=i}^{r-1} \mathbf{D}_{u_j,u_j}} \right) \tag{3}$$

$$= 2 \sum_{e \in E} \sum_{i=1}^r 1$$

$$= 2mr. \tag{4}$$

This conclusion holds since $\mathbf{D}_{u,u} = \sum_{v \in V} \mathbf{A}_{u,v}$. Recall that the derivation of Equation 3 implies the sampling method process of the Laplacian sparsification.

For the proposed Algorithm 1, the probability of sampling a distinct path $p = (u_0, \cdots, u_r)$ can be derived as

$$\Pr(p = (u_0, \cdots, u_r | e, k)) = \Pr((u_{k-1}, u_k) = e) \cdot \Pr(p = (u_0, \cdots, u_r)|(u_{k-1}, u_k) = e)$$

$$= \frac{1}{m} \cdot \prod_{i=1}^{k-1} \frac{\mathbf{A}_{u_i,u_{i-1}}}{\mathbf{D}_{u_i,u_i}} \cdot \prod_{i=k}^{r-1} \frac{\mathbf{A}_{\mathbf{u_i,u_{i+1}}}}{\mathbf{D}_{u_i,u_i}}$$

$$= \frac{1}{m} \cdot \frac{\prod_{i=1}^r \mathbf{A}_{u_{i-1},u_i}}{\left( \prod_{i=1}^{r-1} \mathbf{D}_{u_i,u_i} \right) \cdot \mathbf{A}_{u_{k-1},u_k}}.$$

Since $e$ is sampled uniformly at random, as indicated by the term $\Pr(u_{k-1}, u_k) = e$ above, we now consider the randomness introduced by $k$. $k$ is also sampled uniformly at random. Thus, eliminating $k$ finally yields

$$\Pr(p = (u_0, \cdots, u_r)) = \left( \frac{1}{m} \cdot \frac{\prod_{i=1}^r \mathbf{A}_{u_{i-1},u_i}}{\prod_{i=1}^{r-1} \mathbf{D}_{u_i,u_i}} \right) \left( \frac{1}{r} \cdot \sum_{k=1}^r \frac{1}{\mathbf{A}_{u_{k-1},u_k}} \right)$$

$$= \frac{1}{2mr} w(p) R_{\sup,G_r}(p).$$

Since all the edge weights are considered as 1 on unweighted graphs, the upper bound $R_{\sup,G_r}$ is $2r$ for any $p$ with length $r$. Consequently, the probability of sampling a distinct path $p$ is proportional to $w(p)$. Follow Equation 4, the summation of $w(p)$ for any length $r$ is $m$. We can execute the Monte-Carlo sampling $M$ times to generate $M$ path with length $r$ and weight $m/M$ to generate the unbiased approximation of $\mathbf{D}(\mathbf{D}^{-1}\mathbf{A})^r$.

Cheng et al. (2015) have proven that there exists an algorithm that employs the process outlined above to generate an $\varepsilon$-sparsifier of $\mathbf{D}(\mathbf{D}^{-1}\mathbf{A})^r$ within $M = O(n \log n/\varepsilon^2)$ edges with the probability $1 - 1/n$ at least. Our final objective is to approximate the $\mathbf{D}^{-1/2} \cdot \mathbf{D} \left( \sum_{k=0}^K w_k \left( \mathbf{D}^{-1}\mathbf{A} \right)^k \right) \mathbf{D}^{-1/2}$ proposed in Equation 1.

In general, all of the sampled edges $(u_0, u_r)$, representing path $p = (u_0, ..., u_r)$, should be multiplied by a correction coefficient of $d_{u_0}^{-1/2} d_{u_r}^{-1/2}$, intuitively.

For random walk polynomials with static coefficients, such as in SLSGC, one effective method to approximate the middle term is to distribute all the edges with the probability proportional to $|w_i|$ for

each path length $i$. Hence, we can first pick the length $i$ with the probability $\Pr\{r = i\} = |w_i|/\|\mathbf{w}\|_1$, then sample the edge and correct the edge weight with $d_{u_0}^{-1/2} d_{v_0}^{-1/2} \text{sgn}(w_i) \|\mathbf{w}\|_1$ to compensate for the rescaling of the probability. This process intuitively maintains the unbiased approximation and the Laplacian sparsification properties.

For random walk polynomials with learnable coefficients, the models are required to generate the correct hop-independent derivative of the $w_i$. This limitation prevents us from directly sampling with $w_k$. Instead, we may sample the graph hop-by-hop, meaning that for each random walk length $i$, we independently generate the sparsifier of graph $\mathbf{D}(\mathbf{D}^{-1}\mathbf{A})^i$ and stack them with corresponding coefficients $w_i$. To maintain the property of being an $\varepsilon-$sparsifier of the given graph, one sufficient condition is that all the generated sparsifiers are $\varepsilon$-sparsifiers. This method increases the required number of generated edges to $O(Kn \log n/\varepsilon^2)$, and the probability decreases to $(1 - 1/n)^K \geq 1 - K/n$. Since $K$ is a predefined constant and $K \ll n$, the complexity does not change significantly.

### A.1.2 PROOF OF THEOREM 4.3

We can begin by considering the probability of the sampling method selecting a distinct path on the graph. Assume the desired path is $p = (u_0, \cdots, u_r)$ and the first selected node is $u$, we can derive the probability as follows:

$$\Pr(p = (u_0, \cdots, u_r)) = \Pr(u = u_0) \cdots \Pr(p = (u_0, \cdots, u_r)|u = u_0)$$

$$= \frac{\mathbf{D}_{u_0,u_0}}{\sum_{v \in V_c} \mathbf{D}_{v,v}} \cdot \prod_{i=0}^{r-1} \frac{\mathbf{A}_{u_i,u_{i+1}}}{\mathbf{D}_{u_i,u_i}}$$

$$= \frac{1}{\sum_{v \in V_c} \mathbf{D}_{v,v}} \cdot \frac{\prod_{i=0}^{r-1} \mathbf{A}_{u_i,u_{i+1}}}{\prod_{i=1}^{r-1} \mathbf{D}_{u_i,u_i}}$$

$$= \frac{w(p)}{\sum_{v \in V_c} \mathbf{D}_{v,v}},$$

where $Vc$ is the set of nodes where the start of random walks are selected from.

From the derivation of Equation 4, we can conclude that the summation of weights of all the paths starting from node $u$ can be divided into $\mathbf{D}_{u,u}$ series, where each series starts with one of the $\mathbf{D}_{u,u}$ edges incident with node $u$. Since the summation of all the paths with one distinct $k$ satisfying $(u_k, u_{k+1}) = e$ is 1, we have the summation of weights of all the paths starting from $u$ is $\mathbf{D}_{u,u}$. This means the probability of sampling a distinct path is proportional to its weight.

The entry $\left(\mathbf{D}(\mathbf{D}^{-1}\mathbf{A})^r\right)_{i,j}$ represents the combination of the weights of all the path $p = (u_0 = i, u_1, \cdots, u_{r-1}, u_r = j)$. For each sampled path, we can correct it by multiplying it with $\frac{1}{M} \sum_{v \in V_c} \mathbf{D}_{v,v}$ to obtain an unbiased approximation of the weight of each path. The union of the paths will inevitably generate an unbiased approximation of the corresponding rows of $Vc$ in $\mathbf{D}(\mathbf{D}^{-1}A)^r$. Thus, the final weight generated for the path $p = (u_0, \cdots, u_r)$ is

$$\frac{d_{u_0}^{-1/2} d_{u_r}^{-1/2}}{M} \left( \sum_{v \in V_c} \mathbf{D}_{v,v} \right).$$

### A.2 EXAMPLE PSEUDO-CODES OF THE ENTANGLED

### A.2.1 APPNP

This section presents the pseudo-code of Laplacian sparsification entangled APPNP in Algorithm 4. Note that the hop weight matrix $\mathbf{W}$ is not learnable. $\mathbf{W}$ is predefined as $\mathbf{W}_{0,i} = \alpha(1 - \alpha)^i, i \neq K$ and $\mathbf{W}_{0,K} = (1 - \alpha)^K$, where $\alpha$ is a hyper-parameter.

### A.2.2 GLSGC AND GPRGNN-LS

This section presents the pseudo-code of constructing a Laplacian sparsified random walk matrix polynomial with learnable coefficients in Algorithm 5 and Laplacian sparsification entangled GPRGNN in Algorithm 6. Note that the hop weight matrix $\mathbf{W}$ is learnable. Please follow GPRGNN (Chien et al., 2021) for more details of the initialization of $\mathbf{W}$.

---

**Algorithm 4:** APPNP with Laplacian Sparsification

---

**Input:** Node embeddings $\mathbf{X}$, training status $T$.
**Input: Model Saved:** Vertice set $V$, edge set $E$, degrees $\mathbf{d}$, maximum neighbor hop $K$,
      sampling number $M$, hop weight matrix $\mathbf{W}$.
**Output:** Processed node embeddings $\widetilde{\mathbf{X}}$

1   $\mathbf{X} \leftarrow \text{Linear}(\mathbf{X})$
2   **if** $T$ **then**
3      $\widetilde{E} \leftarrow \text{SLSGC}(\mathbf{W}_0)$
4      $\widetilde{\mathbf{X}} \leftarrow$ A round of message passing of $\mathbf{X}$ on edge set $\widetilde{E}$
5   **else**
6      $\widetilde{\mathbf{X}} \leftarrow \mathbf{W}_{0,0}\mathbf{X}$
7      **for** $i$ from 1 to $K$ **do**
8          $\mathbf{X} \leftarrow$ A round of message passing of $\mathbf{X}$ on edge set $G.E$ with weights $\mathbf{D}^{-1/2}\mathbf{A}\mathbf{D}^{-1/2}$
9          $\widetilde{\mathbf{X}} \leftarrow \widetilde{\mathbf{X}} + \mathbf{W}_{0,i}\mathbf{X}$
10 # Possibly move ahead.
11 $\widetilde{\mathbf{X}} \leftarrow \text{Linear}(\widetilde{\mathbf{X}})$
12 **return** $\widetilde{\mathbf{X}}$

---

**Algorithm 5:** General Laplacian Sparsified Graph Construction (GLSGC)

---

**Input:** Hop weights $\mathbf{w}$.
**Input: Model Saved:** Vertice set $V$, edge set $E$, degrees $\mathbf{d}$, maximum neighbor hop $K$,
      layer-wise sampling number $M$.
**Output:** Weighted edge set $\widetilde{E}$ after laplacian sparsification.

1   $m, \widetilde{E} \leftarrow |E|, \emptyset$
2   **for** each $v \in V$ **do**
3      $\widetilde{E} \leftarrow \widetilde{E} \cup ((v,v), \mathbf{w}_0)$
4   **for** $k$ from 1 to $K$ **do**
5      **for** $j$ from 1 to $M$ **do**
6          $(u,v) \leftarrow \text{Edge\_Sampling}(E, k)$
7          $\widetilde{E} \leftarrow \widetilde{E} \cup \left( (u,v), \mathbf{w}_k d_u^{-1/2} d_v^{-1/2} \cdot \frac{m}{M} \right)$
8 **return** $\widetilde{E}$

---

### A.2.3   Node-wise Laplacian Sampling

In this section, Algorithm 7 presents the pseudo-code of the node-wise Laplacian sampling algorithm for semi-supervised tasks.

### A.3   Dataset Details

Here we list the detailed information of used datasets in the experiment. All the information we proposed here refers to the original version of data collected by PyG. All the graphs are not converted to the undirected graph and added self-loops, which are precomputed and saved.

### A.4   Experiment Details

### A.4.1   Configurations

Here we list the detailed information on the experimental platform and the environment we deployed.

- Operating System: Red Hat Linux Server release 7.9 (Maipo).

- CPU: Intel(R) Xeon(R) Gold 8358 64C@2.6GHz.

---

**Algorithm 6:** GPRGNN with Laplacian Sparsification

---

**Input:** Node embeddings $\mathbf{X}$, training status $T$.
**Input: Model Saved:** Vertice set $V$, edge set $E$, degrees $\mathbf{d}$, maximum neighbor hop $K$,
      sampling number $M$, hop weight matrix $\mathbf{W}$.
**Output:** Processed node embeddings $\widetilde{\mathbf{X}}$

1   $\mathbf{X} \leftarrow \text{Linear}(\mathbf{X})$
2   **if** $T$ **then**
3      $\widetilde{E} \leftarrow \text{GLSGC}(\mathbf{W}_0)$
4      $\widetilde{\mathbf{X}} \leftarrow$ A round of message passing of $\mathbf{X}$ on edge set $\widetilde{E}$
5   **else**
6      $\widetilde{\mathbf{X}} \leftarrow \mathbf{W}_{0,0}\mathbf{X}$
7      **for** $i$ from 1 to $K$ **do**
8          $\mathbf{X} \leftarrow$ A round of message passing of $\mathbf{X}$ on edge set $G.E$ with weights $\mathbf{D}^{-1/2}\mathbf{A}\mathbf{D}^{-1/2}$
9          $\widetilde{\mathbf{X}} \leftarrow \widetilde{\mathbf{X}} + \mathbf{W}_{0,i}\mathbf{X}$
10   # Possibly move ahead
11   $\widetilde{\mathbf{X}} \leftarrow \text{Linear}(\widetilde{\mathbf{X}})$
12   **return** $\widetilde{\mathbf{X}}$

---

**Algorithm 7:** Node-Wise Laplacian Sampling

---

**Input:** Hop weights $\mathbf{w}$, batch $B$
**Input: Model Saved:** Vertice set $V$, edge set $E$, degrees $\mathbf{d}$, maximum neighbor hop $K$,
      sampling number $M$.
**Output:** Weighted edge set $\widetilde{E}$ after random walk sampling.

1   $s \leftarrow \sum_{u \in B} \mathbf{d}_u$
2   $\widetilde{E} \leftarrow \emptyset$
3   **for** $k$ from 1 to $K$ **do**
4      **for** $j$ from 1 to $M$ **do**
5          $u \leftarrow$ sample a node in $B$ from distribution $\Pr\{u = u_0\} = \mathbf{d}_{\mathbf{u_0}}/s$
6          $v \leftarrow$ the end of a random walk on $E$ (i.e. graph $G$), starting from $u$ with length $k$
7          $\widetilde{E} \leftarrow \widetilde{E} \cup \left( (u,v), \mathbf{w}_k d_u^{-1/2} d_v^{-1/2} \cdot \frac{s}{M} \right)$
8   **return** $\widetilde{E}$

---

- GPU: NVIDIA A100 40GB PCIe.
  GPU: NVIDIA A100 80GB PCIe for the experiment on Ogbn-papers100M.
- GPU Driver: 525.125.06.
- RAM: 512G.
- Python: 3.10.6.
- CUDA toolkit: 11.3.
- Pytorch: 1.12.1.
- Pytorch-geometric: 2.1.0.

A.4.2   DETAILS FOR SMALL-SCALE EXPERIMENTS.

For all the datasets involved in Table 1, we conduct a full-supervised node classification experiment. Except for Twitch-de, all the datasets are randomly divided into 10 splits with the commonly adapted training/validation/test proportions of 60%/20%/20%. For the Twitch-de, we use the default 5 splits proposed by PyG and propose the average performance of all the datasets.

We limit the hidden size to 64 and the layers of MLP to 2 for a relatively fair comparison among all the tested models. We propose the best hyperparameters of our Laplacian sparsification extended models in Table 4 and 5 for full reproducibility.

Table 3: Detailed information about used datasets.

| Dataset | Nodes | Edges | Features | Classes | $\mathcal{H}(\mathcal{G})$ | scale |
|---|---|---|---|---|---|---|
| Cora | 2,708 | 5,278 | 1,433 | 7 | 0.810 | small |
| Citeseer | 3,327 | 4,552 | 3,703 | 6 | 0.736 | small |
| PubMed | 19,717 | 44,324 | 500 | 3 | 0.802 | small |
| Actor | 7,600 | 30,019 | 932 | 5 | 0.219 | small |
| Wisconsin | 251 | 515 | 1,703 | 5 | 0.196 | small |
| Cornell | 183 | 298 | 1,703 | 5 | 0.305 | small |
| Texas | 183 | 325 | 1,703 | 5 | 0.108 | small |
| Photo | 7,650 | 119,081 | 745 | 8 | 0.827 | small |
| Computers | 13,752 | 245,861 | 767 | 10 | 0.777 | small |
| Twitch-de | 9,498 | 153,138 | 2,514 | 2 | 0.632 | small |
| Ogbn-arxiv | 169,343 | 1,166,243 | 128 | 40 | 0.655 | medium |
| Twitch-gamers | 168,114 | 6,797,557 | 7 | 2 | 0.554 | medium |
| Penn94 | 41,554 | 1,362,229 | 4,814 | 2 | 0.489 | medium |
| Ogbn-papers100M | 111,059,956 | 1,615,685,872 | 128 | 172 | - | large |

Table 4: The hyper-parameters of APPNP-LS on small-scale datasets.

| Dataset | learning rate | weight decay | dropout=dprate | $\alpha$ | $K$ | ec |
|---|---|---|---|---|---|---|
| Cora | 0.05 | 0.0005 | 0.5 | 0.1 | 5 | 10 |
| Citeseer | 0.01 | 0 | 0.8 | 0.5 | 10 | 10 |
| PubMed | 0.05 | 0.0005 | 0.0 | 0.5 | 10 | 10 |
| Actor | 0.01 | 0.0005 | 0.8 | 0.9 | 2 | 10 |
| Wisconsin | 0.05 | 0.0005 | 0.5 | 0.9 | 10 | 20 |
| Cornell | 0.05 | 0.0005 | 0.5 | 0.9 | 5 | 20 |
| Texas | 0.05 | 0.0005 | 0.8 | 0.9 | 2 | 20 |
| Photo | 0.05 | 0 | 0.5 | 0.5 | 5 | 10 |
| Computers | 0.05 | 0 | 0.2 | 0.1 | 2 | 10 |
| Twitch-de | 0.01 | 0.0005 | 0.5 | 0.1 | 2 | 10 |

### A.4.3  DETAILS FOR LARGE-SCALE EXPERIMENTS

For all the datasets involved in Table 2, we conduct a supervised node classification experiment. Since the proportion of training nodes in Ogbn-papers100M is small, we consider it a semi-supervised task. All the datasets here have the default splits provided by PyG. For Twitch-gamers and Penn94, we propose the average accuracy of all the default splits. For OGB datasets, we repeat the experiment on the single split 5 times with different random seeds and report the average accuracy.

We limit the hidden size to 128 and the layers of MLP to 3 at most for a relatively fair comparison among all the tested models. We propose the best hyperparameters for our Laplacian sparsification extended models in Table 6 and 7 for full reproducibility.

Table 5: The hyper-parameters of GPR-LS on small-scale datasets.

| Dataset | lr=prop_lr | wd=prop_wd | dropout=dprate | $\alpha$ | $K$ | ec |
|---|---|---|---|---|---|---|
| Cora | 0.05 | 0.0005 | 0.8 | 0.9 | 10 | 3 |
| Citeseer | 0.01 | 0 | 0.2 | 0.5 | 10 | 10 |
| PubMed | 0.05 | 0.0005 | 0.2 | 0.5 | 5 | 10 |
| Actor | 0.05 | 0.0005 | 0.5 | 0.5 | 10 | 3 |
| Wisconsin | 0.05 | 0.0005 | 0.8 | 0.9 | 10 | 10 |
| Cornell | 0.05 | 0.0005 | 0.5 | 0.9 | 2 | 1 |
| Texas | 0.05 | 0.0005 | 0.8 | 0.5 | 10 | 10 |
| Photo | 0.05 | 0.0005 | 0.8 | 0.1 | 2 | 10 |
| Computers | 0.05 | 0 | 0.5 | 0.1 | 10 | 10 |
| Twitch-de | 0.01 | 0.0005 | 0.8 | 0.1 | 2 | 20 |

Table 6: The hyper-parameters of APPNP-LS on medium- and large-scale datasets.

| Dataset | learning rate | weight decay | dropout=dprate | $\alpha$ | $K$ | ec |
|---|---|---|---|---|---|---|
| Ogbn-arxiv | 0.01 | 0 | 0 | 0.1 | 5 | 10 |
| Twitch-gamer | 0.01 | 0 | 0 | 0.5 | 2 | 10 |
| Penn94 | 0.01 | 0 | 0.5 | 0.9 | 10 | 10 |
| Ogbn-papers100M | 0.01 | 0 | 0.2 | 0.1 | 2 | - |

Table 7: The hyper-parameters of GPR-LS on medium- and large-scale datasets.

| Dataset | lr=prop_lr | wd=prop_wd | dropout=dprate | $\alpha$ | $K$ | ec |
|---|---|---|---|---|---|---|
| Ogbn-arxiv | 0.01 | 0 | 0.2 | 0.5 | 10 | 10 |
| Twitch-gamer | 0.05 | 0 | 0.5 | 0.1 | 5 | 10 |
| Penn94 | 0.01 | 0 | 0.5 | 0.1 | 2 | 10 |
| Ogbn-papers100M | 0.01 | 0 | 0.2 | 0.9 | 2 | - |

