# OpenReview forum: "Large-Scale Spectral Graph Neural Networks via Laplacian Sparsification"
_ICLR.cc/2024/Conference — Submitted to ICLR 2024_

### Official Review · Reviewer_injV · 2023-10-19

**Soundness:** 2 fair
**Presentation:** 3 good
**Contribution:** 2 fair
**Rating:** 5
**Confidence:** 2

**Summary:**

The authors present a spectral sparsification approach to improve the scalability of spectral graph neural networks, which avoids detaching and enables end-to-end training. The authors also test the efficacy of the spectral sparsification for different datasets, including a very large-scale graph dataset.

**Strengths:**

The experiments cover datasets of different sizes, including very big ones, which is a strong point.

**Weaknesses:**

The theory is a rather simple application of the results of Daniel A. Spielman et al.’s theory. The theory only shows some relations between the original graph and the sparsified graph. However, it does not give any results about the performance of GNNs. The theory is disconnected from the GNN theme of this paper.

Spectral sparsification for GNNs has been used widely in GNNs; the authors seem to ignore all related works that use spectral sparsification in the context of GNNs.

Instead of using spectral sparsification, the paper “Johannes Gasteiger, Stefan Weißenberger, Stephan Günnemann, Diffusion Improves Graph Learning, NeurIPS 2019” uses a thresholding approach for spectral approaches. Can the authors comment on this and provide some comparisons?

The authors should include the computational complexity analysis for both memory and computational time. When taking the spectral sparsification step into account, the proposed approach seems also to require a very large memory footprint.

Numerical comparisons with the detached approach are missing.

The authors may consider comparing against other approaches for scalable GNNs, e.g. Clustered GCNs.

Report standard deviation - the improvement seems rather small; perhaps within the standard deviation.

**Questions:**

See the questions I mentioned in the weaknesses part.

---

> ### Author Response · Authors · 2023-11-18
>
> Thanks for your time and efforts spent on our work and detailed feedback!
>
> **Reply to weaknesses**
>
> - Our model applies a spectral sparsification method to random-matrix polynomials and extends the theorem to include arbitrary (dynamic and learnable) polynomial coefficients.
> Additionally, we introduce a more intuitive method for enhancing efficiency in semi-supervised tasks, which functions as spectral sparsification.
> We acknowledge that the performance of GNN models can be empirical, particularly with the inclusion of a linear layer.
> Previous works like GraphSAINT and Cluster-GNN have provided substantial analysis on estimation and approximation, though not directly on the GNNs' performance.
> Our model's approximation capability has been demonstrated both theoretically and empirically.
>
> - We appreciate your input regarding the focus of most spectral sparsification methods on the original graph.
> Our work, however, concentrates on approximating the equivalent propagation matrix of spectral methods, $L_K=\sum_{k=0}^K w_k\tilde{L}^k$.
> We initially consider that our model differs from these past methods since our target is not the same.
> Recognizing the distinction, we will promptly include these methods in our study for relevant comparisons.
>
> - We appreciate your reminding us of the model GDC.
> The design of GDC is based on the homophily assumption, which may not perform as well on heterophilous datasets compared to popular spectral methods.
> We will incorporate GDC into our paper for a more comprehensive discussion and applicable comparisons.
>
> - We will instantly complement and clarify statements about the time and memory complexity.
> If processed with a single thread, the time complexity of running SLSGC is $O(K n\log n)$, and for GLSGC, it is $O(K^2n\log n)$.
> Besides, our method can be highly parallel processed with GPU.
> The memory consumption, dependent on the number of edges sampled, is $O(n \log n)$ for SLSGC and $O(Kn\log n)$ for GLSGC.
> Our experimental results show that the training time and memory consumption are lower than the base models, and here are some preliminary results for preview.
> The detailed version will be soon updated in our paper.
>
> | time per epoch (ms) | Cora | Computers | Twitch-gamer | Penn94 |
> | ------------------- | ---- | --------- | ------------ | ------ |
> | APPNP               | 6.50 | 7.59      | 19.99        | 13.75  |
> | APPNP-LS            | 5.58 | 5.83      | 16.73        | 12.42  |
>
> | GPU memory consumption (GB) | Cora  | Computers | Twitch-gamer | Penn94 |
> | --------------------------- | ----- | --------- | ------------ | ------ |
> | APPNP                       | 0.034 | 0.144     | 0.976        | 1.725  |
> | APPNP-LS                    | 0.034 | 0.127     | 0.845        | 1.821  |
>
> - Detaching models like SGC often show lower performance due to their simpler architecture, in comparison with GCN.
> Another defect of detaching models is that the graph propagation of raw features is inapplicable when the dimension of raw features is high.
> We will conduct and complete related experiments to provide a thorough comparison, and here are some experimental results for preview.
>
> |          | Cora           | Cite.          | PubM.          | Actor          | Wisc.          | Corn.          | Texas          |
> | -------- | -------------- | -------------- | -------------- | -------------- | -------------- | -------------- | -------------- |
> | GPR      | 88.80$\pm$1.17 | 81.57$\pm$0.82 | 90.98$\pm$0.25 | 40.55$\pm$0.96 | 91.88$\pm$2.00 | 89.84$\pm$1.80 | 92.78$\pm$2.30 |
> | GPR-LS   | 89.31$\pm$1.07 | 81.65$\pm$0.53 | 90.95$\pm$0.37 | 41.32$\pm$0.83 | 93.63$\pm$2.88 | 91.14$\pm$1.15 | 92.62$\pm$1.48 |
> | GPRdet   | 88.13$\pm$0.85 | 80.96$\pm$1.05 | 90.96$\pm$0.29 | 40.10$\pm$0.61 | 88.88$\pm$2.75 | 89.83$\pm$2.46 | 85.08$\pm$2.62 |
>
> - We will add scalable methods like Cluster-GCN and FastGCN to our paper for a more comprehensive comparison with relevant works.
> Here are some experimental results for preview.

---

> > ### Author Response · Authors · 2023-11-18
> >
> > |            | Cora      | Cite.     | PubM.     | Actor     | Wisc.     | Corn.     | Texas     | Photo    | Comp.     | Twit.     |
> > | ---------- | --------- | --------- | --------- | --------- | --------- | --------- | --------- | -------- | --------- | --------- |
> > | GPR        | 88.80     | 81.57     | **90.98** | 40.55     | 91.88     | 89.84     | **92.78** | 95.10    | 89.69     | 73.90     |
> > | GPR-LS     | **89.31** | **81.65** | 90.95     | **41.82** | **93.63** | **91.14** | 92.62     | **95.3** | **90.47** | **73.49** |
> > | $\Delta$   | +0.51     | +0.08     | -0.03     | +1.27     | +1.75     | +1.30     | -0.16     | +0.20    | +0.78     | -0.42     |
> > | ClusterGCN | 87.45     | 79.66     | 86.52     | 29.66     | 61.88     | 56.72     | 65.08     | 93.17    | 87.11     | 68.69     |
> > | GraphSAGE  | 87.95     | 80.06     | 88.82     | 39.27     | 90.37     | 85.57     | 86.39     | 94.44    | 89.29     | 67.28     |
> >
> > - A 95\% confidence interval will be included in our experimental results.
> > As an approximate method, our model's performance is closely linked to the base method.
> > While many methods like Cluster-GCN, GraphSAGE, GraphSAINT, LADIES, and GNNAutoScale experience performance loss, particularly as benchmarks scale, our model maintains, and in some cases surpasses, the performance of strong base models like GPR-GNN.
> > This underscores the superiority of our approach, balancing efficiency, performance, and applicability effectively.
> >
> > We are grateful for the thorough feedback and remain open to further discussion and clarification on any aspect of our work.

---

> > > ### Comment · Reviewer_injV · 2023-11-22
> > > **Response to Authors**
> > >
> > > Dear authors,
> > >
> > > Thank you for taking my comments seriously and providing a thorough rebuttal. Overall, I still feel the novelty of the work seems rather limited and I have to lower my confidence in my rating of your paper. I will consider your paper again after seeing the revised version.

---

### Official Review · Reviewer_FyWK · 2023-10-28

**Soundness:** 3 good
**Presentation:** 2 fair
**Contribution:** 2 fair
**Rating:** 5
**Confidence:** 3

**Summary:**

This paper proposes a method for sparsifying polynomials of graph Laplacians by sampling random edges from some random walk shift operators. The goal is to speed up the applications of Laplacian polynomials in spectral GNNs. Experiments show that the approach sometimes improves performance on well known benchmarks.

**Strengths:**

The method is clearly explained. The method is analyzed theoretically, and the experiments indicate that the sparsification method often improves out-of-the-box GNN methods.

**Weaknesses:**

First, the method is mostly an application of a well known Laplacian polynomial sparsification method.

The main problem with the paper at its current form is that important related methods are not cited and compared against. It is hence difficult to judge the paper and understand where the proposed method sits with respect to other methods.

Let me write a partial list of missing papers that need to be compared against.

**Papers about subsampling graphs, motivated by scalability:**

J. Chen, T. Ma, and C. Xiao. FastGCN: Fast learning with graph convolutional networks via importance sampling. In International Conference on Learning Representations, 2018.

W.-L. Chiang, X. Liu, S. Si, Y. Li, S. Bengio, and C.-J. Hsieh. Cluster-gcn: An efficient algorithm for training deep and large graph convolutional networks. In Proceedings of the 25th ACM SIGKDD International Conference on Knowledge Discovery and Data Mining,

There are other papers along these lines. The authors need to survey the literature.

**Paper about precomputing diffusion before training:**

E Rossi, F Frasca, B Chamberlain, D Eynard, M Bronstein, F Monti
. SIGN: Scalable Inception Graph Neural Networks

**Transferability/stability to subsampling:**

In addition, there are many theoretical papers about the stability/transferability of GNNs with respect to graph subsampling. The first important three papers are

N Keriven, A Bietti, S Vaiter. Convergence and Stability of Graph Convolutional Networks on Large Random Graphs

R Levie, W Huang, L Bucci, M Bronstein, G Kutyniok. Transferability of spectral graph convolutional neural networks

 L Ruiz, L Chamon, A Ribeiro. Graphon neural networks and the transferability of graph neural networks

There are many more papers along this line. The authors should survey subsequent papers by the authors of the above three papers and others.

To get an $\epsilon$ approximation, these papers seem to need only $O(1/\epsilon^2)$ sampled nodes, which for dense graphs amounts to $O(1/\epsilon^4)$ edges. This is independent of the degree of the graph, while in your paper the number of edges is linear in the degree of the graph. You need to thoroughly compare your results to these papers. Is the different dependency on the order a result of using a different norm? If so, explain how to compare between the results by converting to the same norm. If your result fundamentally has worse dependency on the degree of the graph, you need to explain what you gain on account of this worse dependency. Without a thorough comparison to past works it is difficult to gauge the contribution of your paper.

There is also a whole field about matrix sketching methods which is relevant. One classic approach is to sample the rows of a large matrix randomly to reduce complexity.


The paper should compare against the above methods, and explain what is novel about the proposed approach with respect to past methods, what the proposed method improves, and what are the shortcomings of the new method with respect to past methods. If this comparison is long, a short version can be written in the main paper, and an extended section can be written as an appendix.

Moreover, note that in [Spectral sparsification of random-walk matrix polynomials
] they want to approximate the polynomial of the matrix itself, not the application of the polynomial of the matrix on the signal. In your work you are only interested in applying the polynomial filter on the signal. For this, you have simple efficient implementations if the graph is sparse: you apply $L$ on $L^kX$ by induction, $k=0,\ldots,K$ to compute all monomials filters $L^kX$ applied on $X$ in linear time in the number of edges (times the power $K$). You need to explain better what your method improves with respect also to this direct method.


Appendix A.1 about the proof of Theorem 3.2 is not clear, and does not seem to have a  rigorous proof. It would be better to write a proof inside a traditional proof environment. You should clearly state in what sense you extend the proof of (Cheng et al., 2015), and what is taken from (Cheng et al., 2015).

**Questions:**

**Detailed (and minor) comments:**

Page 2, first paragraph, another big problem with the precomputation of L^k X is that the network can only have one layer. You cannot precompute the powers of the Laplacian on hidden layers.

Contribution:

“keeping the number of non-zeros within an acceptable range” Nonzeros of what? Write more explicitly.


“Scalable method designing” “which is the first work tackling the scalability issue of spectral GNNs to the best of our knowledge” There are many papers that deal with that, including papers that you cite. Please say that you propose a new way for scalability.


Page 3, MOTIVATION: Use consistent notation. You sometimes use small $x$ and sometimes large $X$ for the input signal. This section mainly repeats things that were already written before. Especially the last paragraph.


Definition 3.1: correct “semi-definite” to “positive semi-definite.”


First line in Page 4 - you forgot period: “eigenvalues are in close correspondence.”

Equation (2): the first approximation is wrong. For example, take $K=1$ and $w_1=1$, and note that $P$ does not approximate $L$. Do you mean that there is a choice of DIFFERENT coefficients $w’_k$ for the polynomial in $P$ that gives an approximation? In that case, you can get an exact equality.
Also, the powers of $L$ should be small $k$.

Two lines below (2): change “desiring matrix” to “desired matrix”.

Theorem 3.2 is formulated in a confusing way. Writing ``we can construct an $\epsilon$ sparsifier’’ sounds like an existence claim, but what you are trying to say is that in probability $1-K/n$ Algorithm 1 gives an $\epsilon$ sparsifier.

There are other papers about subsampling graphs that get rid of the dependency on the degree of the graph. For example, see Theorem, 1 in [N Keriven, A Bietti, S Vaiter. Convergence and Stability of Graph Convolutional Networks on Large Random Graphs]. There, to get an $\epsilon$ error you need to sample $O(1/\epsilon^2)$ nodes, which is independent of the size of the graph. How do you explain the slower asymptotics in your results? What do you gain with respect to the past analyses on account of slower asymptotics? You need to discuss this in detail.

Why would computing random edges make things faster for sparse L? If the number of edges in L is O(n), then already computing $L_K$ takes O(Kn) operations. In your method you need $O(K^2n)$ operations to construct the whole polynomial.
Perhaps your method is only useful for dense graphs? More accurately, when the number of edges is >> the number of nodes? However, it is well known that you can approximate such dense graph shift operators via Monte Carlo sampling the nodes, as I wrote above. Please motivate your method accordingly.  For example, you can compare the complexity to methods that directly apply $L$ on $X$ as many times as needed for the polynomial, assuming that the number of edges is $m=O(n^a)$ where $n$ is the number of nodes and $a$ is between 1 and 2.


Section 4: please define effective resistance.

Page 7: please add the reference “This sparsifier can be further reduced to O(n log n/ε2 ) by the existing works []”

“Note that the proposed bound is much tighter than what is practically required. In practice, the sampling number can be set rather small to achieve the desired performance”  - you mean, the proposed bound is much higher than…?

---

> ### Comment · Reviewer_injV · 2023-11-11
> **N Keriven's paper seems irrelevant to this work**
>
> This paper uses Spielman et al.'s sparsification for general graphs. How is the paper "N Keriven, A Bietti, S Vaiter. Convergence and Stability of Graph Convolutional Networks on Large Random Graphs" related? Can Reviewer FyWK or N Keriven comment on this and make it more clear? I thank the reviewer of N Keriven in advance.

---

> ### Comment · Reviewer_FyWK · 2023-11-11
> **N Keriven's paper is related**
>
> The current paper shows that you can approximate GNNs by randomly sampling edges. N Keriven shows that you can approximate GNNs by randomly sampling nodes. These are two related approaches, both related to scalability of GNNs by subsampling graphs. This is why the current approach should be compared to Keriven et al. Practitioners would want to choose between these two approaches, and they would like to see a comparison between them. What is the advantage in sampling edges over sampling nodes?
>
> Note that there is a full asymptotic analysis in Keriven et al, so the rate of approximation should be compared to the current paper.

---

> ### Comment · Reviewer_injV · 2023-11-11
> **Thanks but I feel N Keriven’s paper is totally irrelevant**
>
> The current paper uses effective resistance for sparsification, which is quite generic and does not need any special assumption. N Keriven’s paper is based on many assumptions and does not consistent with the current work. How can N Keriven’s paper related and the first one in the three important papers? I need more help to understand this.

---

> > ### Comment · Reviewer_FyWK · 2023-11-11
> >
> > I wrote above how it is relevant. Both papers deal with subsampling graphs in GNNs. If there are special assumptions in Keriven's paper this should be written as part of the comparison in the paper. This is what comparisons are for.

---

> > > ### Comment · Reviewer_injV · 2023-11-11
> > > **Thanks**
> > >
> > > Thanks. However, N Keriven's paper does not include Spielman et al's work as related works.

---

> ### Comment · Reviewer_FyWK · 2023-11-11
>
> Practitioners reading this paper, who want to use a scalability method, would want to know what are the advantages and disadvantages in sampling edges vs sampling nodes. If the paper proposes a new sampling method - sampling edges - it should compare against past sampling methods. Especially since past works developed approximation rates, so you can compare to the approximation rates of the current paper. This is the standard way of writing papers.
>
> Regarding papers that Keriven cited, I don't see how this is relevant. We are reviewing the current paper. Moreover, Keriven did not have a gnn paper about sampling edges to compare against. The current paper does have a gnn paper about sampling nodes to compare against.

---

> > ### Author Response · Authors · 2023-11-11
> > **Thanks for discussion**
> >
> > Thank you all for engaging in the discussion regarding the relevance of the paper "N Keriven, A Bietti, S Vaiter. Convergence and Stability of Graph Convolutional Networks on Large Random Graphs" to our paper.
> >
> > The mentioned paper highlights the practicality of node-based subsampling methods in scaling GNNs to a larger scenario. Our proposed methods focus on preserving the spectral properties of the equivalent dense propagation matrix, $L_K=\sum_{k=0}^K w_k\tilde{L}^k$ (i.e. spectral filtered Laplacian), through edge sampling.
> >
> > Node-based sampling methods are prevalent in practical use. However, node-based methods like subgraph sampling methods explicitly lose the information of graph structure. The paper of N Keriven delves into the convergence and stability of GNNs with sampling under the strong assumption of random graphs, while our paper focuses on the graph structure connection between the sampled graph and the desired graph (here the $L_K$).
> >
> > We acknowledge that the paper mentioned by the reviewer FyWK indeed intersects with our work in the realm of scalability. It is noteworthy that many node-based scalability approaches do not work well in practice or have difficulty in the application when entangled with spectral GNNs. We plan to reference this paper and make some applicable comparisons in our work later.

---

> > > ### Comment · Reviewer_FyWK · 2023-11-12
> > > **The graph that you sample need not be random**
> > >
> > > Note that there is no special requirement for the full graph G in N Keriven et al to be random. Only the sampled graph is random. You can take the graphon to be the induced graphon from the original graph G that you want to sample (or some approximation of that), and then the sampled graph from this induced graphon is identical to a sampled graph from G.

---

> ### Author Response · Authors · 2023-11-18
>
> Thanks for your time and efforts spent on our work and detailed feedback!
>
> **Reply to weaknesses.**
>
> - Our model employs a spectral sparsification method for the random-matrix polynomial and extends the theorem to include arbitrary polynomial coefficients, both static and dynamic.
> Furthermore, we propose an intuitive method to enhance efficiency in semi-supervised tasks, functioning as spectral sparsification.
>
> - We are thankful for the comprehensive recommendations regarding related works.
> While we recognize the importance of these scalability-motivated subsampling efforts, we note that most existing scalable GNN frameworks, are based on message-passing principles, such as GCN, GCNII, and GAT.
> These methods require only one-hop neighbor information at each propagation layer, differing from spectral methods.
> Scalable methods like GraphSAGE, LADIES, and GNNAutoScale are not compatible with spectral approaches.
> Similarly, subgraph sampling methods like Cluster-GCN and GraphSAINT do not adequately preserve the spectral properties of the original graph, making them unsuitable for spectral methods.
> We initially perceived a gap between methods designed for GCN-like networks and our work, potentially leading to unfair comparisons.
> Despite these, a more extensive survey and comparison with existing scalable GNNs based on subsampling will be conducted.
> Here are some experimental results for preview.
>
> |            | Cora      | Cite.     | PubM.     | Actor     | Wisc.     | Corn.     | Texas     | Photo    | Comp.     | Twit.     |
> | ---------- | --------- | --------- | --------- | --------- | --------- | --------- | --------- | -------- | --------- | --------- |
> | GPR        | 88.80     | 81.57     | **90.98** | 40.55     | 91.88     | 89.84     | **92.78** | 95.10    | 89.69     | 73.90     |
> | GPR-LS     | **89.31** | **81.65** | 90.95     | **41.82** | **93.63** | **91.14** | 92.62     | **95.3** | **90.47** | **73.49** |
> | $\Delta$   | +0.51     | +0.08     | -0.03     | +1.27     | +1.75     | +1.30     | -0.16     | +0.20    | +0.78     | -0.42     |
> | ClusterGCN | 87.45     | 79.66     | 86.52     | 29.66     | 61.88     | 56.72     | 65.08     | 93.17    | 87.11     | 68.69     |
> | GraphSAGE  | 87.95     | 80.06     | 88.82     | 39.27     | 90.37     | 85.57     | 86.39     | 94.44    | 89.29     | 67.28     |
>
> - Our sparsification targets the equivalent propagation matrix of spectral methods, $L_K=\sum_{k=0}^K w_kL^K$, a dense matrix.
> Unlike previous works that focus on making the original graph sparser, our method aims to transform a dense matrix into a sparse one suitable for GNN training.
> The superiority of spectral methods in handling complex benchmarks, like heterophilous graphs, is evident both theoretically and empirically.
> While our methods are primarily designed for spectral approaches, a thorough comparison with classic scalable methods will be included.
>
> - Matrix sketching methods are effective with real-world graphs, typically sparse.
> However, sampling rows of $L_K$ results in a dense submatrix, still unsuitable for our purposes.
> We plan to conduct an extensive review of these techniques and incorporate comparisons in our upcoming paper.
>
> - In our paper, one of our main ideas is to enhance the theorem proposed in [Spectral sparsification of random-walk matrix polynomials], and then sparsify the equivalent propagation matrix of spectral methods, i.e. $L_K=\sum_{k=0}^K w_k\tilde{L}^k$.
> We turn the polynomial filter operating on the signal into the corresponding format, i.e. the polynomial of the random-walk matrix, and thus make the training of spectral methods on large graphs applicable.
> The mentioned direct method has two defects, which are all reflected in our empirical results: 1) high-dimensional original features can cause GPU memory overload during initial graph propagation (e.g.,
> $LX$ in dataset Penn94 on SGC), and 2) extremely large graphs like Ogbn-papers100M exceed GPU capacity, preventing graph propagation on hidden layers.
>
> - We appreciate the reviewer's advice on improving our writing and theoretical proofs.
> We commit to a thorough examination and enhancement of our theoretical framework.
>
> **Reply to questions.**
> - Yes, and that is how most spectral methods like BernNet improve their scalability.
> In our paper, we refer to these as "detached methods" and highlight their limitations.
> One of our contributions is enabling the spectral filtering operations applicable on hidden layers, even on large graphs.
>
> - We appreciate your scrutiny and will correct our paper accordingly.
> The mentioned "non-zeros of the approximated $L_K$" will be more clearly defined.

---

> > ### Author Response · Authors · 2023-11-18
> >
> > - Thank you for the correction.
> > We will refine the scope of our work and reorganize our writing to ensure accuracy and rigor.
> >
> > - We will correct all the typos promptly.
> > Additionally, we will streamline our wording to convey our motivation concisely.
> >
> > - The omission of "positive" will be addressed and added to our paper immediately.
> >
> > - We will add the missing period and make other necessary typographical corrections.
> >
> > - The error concerning the approximation (small $k$ instead of $K$) will be rectified instantly.
> >
> > - Any additional typos identified will be corrected without delay.
> >
> > - To alleviate any confusion, we will rephrase our statement to more precisely indicate that "The algorithm provides an $\varepsilon$ sparsifier with a high probability."
> >
> > - As highlighted earlier, the primary distinction between our method and past subsampling approaches lies in the objective of approximation.
> > While existing methods focus on approximating the original graph to maintain performance in GCN or other message-passing methods, our approach aims to approximate $L_K=\sum_{k=0}^K w_k\tilde{L}^k$, thereby enhancing the scalability of more potent spectral methods.
> > The original graph could be sparse, while the number of nonzeros in $L_K$  grows exponentially when $K$ grows.
> > The challenge of scalability is more pronounced in spectral methods due to multi-hop neighbor aggregation, justifying the additional effort in approximation.
> >
> > - Computing the random edges does not always make things faster for $L$, especially for the sparse one.
> > One of our motivations is the computational difficulty of the whole graph propagation (i.e. $LX$) in many large-scale scenarios.
> > This issue is particularly acute in cases of precomputation with high-dimensional raw features or during propagation in hidden layers.
> > **Calculating $LX$ iteratively** limits the applicability of several existing scalable methods, including GraphSAGE or GNNAutoScale.
> > Hence, we propose to use laplacian sparsification to approximate $L_K$, a dense matrix.
> > This approximation enables us to store $L_KX$ and transform the $L_KX$ computation into one-time propagation, without calculating $LX$ iteratively.
> > As the neighborhood explosion phenomenon exists in real-world graphs, Monte-Carlo sampling the nodes still involves an exponentially growing number of neighbors when $K$ grows, which is still not operatable in datasets like Ogbn-papers100M.
> > GPR-GNN is the well-developed spectral method that directly applies $L$ on $X$ for constructing the polynomial. We will exert such theoretical analysis in comparing the base models and our entangled model.
> >
> > - We acknowledge the recommendation and will provide a clear definition of relevant terms in the appropriate sections of our paper.
> >
> > - The missing reference will be added promptly to ensure completeness and accuracy.
> >
> > - Our intent is to demonstrate that the theoretical bound of required edges is significantly higher than what is actually necessary for GNN training, without compromising performance. This claim is substantiated by our empirical results.
> >
> > We are grateful for your detailed feedback and remain open to further discussion on any aspect of our work that may require additional clarification or interest.

---

> ### Comment · Reviewer_FyWK · 2023-11-22
>
> Unfortunately, since the authors did not submit a revised version of the paper, and since an extensive revision is required, I cannot reevaluate the paper. As I understand the review process, a revised version could/should have been uploaded as part of the discussion period.
>
> I hope to see a revised version of this paper published in the future.

---

### Official Review · Reviewer_jUvN · 2023-11-01

**Soundness:** 2 fair
**Presentation:** 2 fair
**Contribution:** 2 fair
**Rating:** 3
**Confidence:** 5

**Summary:**

This work leverages prior random-walk-based spectral sparsification for improving the scalability of GNNs. Compared with prior works, the proposed framework allows for end-to-end training of GNNs. This framework allows approximating the equivalent propagation matrix of Laplacian filters, making it compatible with existing scalable techniques. In addition, rigorous mathematical proofs have been provided to support the proposed method.

**Strengths:**

1. It is an interesting idea to leverage spectral sparsification for improving the scalability of the GNN training phase.
2. Rigorous mathematical proofs have been provided to support the proposed method.

**Weaknesses:**

1. It would be helpful if there were more detailed discussions and explanations when claiming that "our models show a slight performance advantage over the corresponding base models" on the heterophilous datasets proposed by Lim et al. (2021), and when discussing "approximating the desired complex filters tailored to the heterophilous graphs."
2. The theoretical analysis in this work is mostly based on the prior work of "Dehua Cheng, Yu Cheng, Yan Liu, Richard Peng, and Shang-Hua Teng. Spectral sparsification of random-walk matrix polynomials. CoRR, abs/1502.03496, 2015," and is a bit incremental.
3. The proposed framework has many hyperparameters, which may make it impractical for use in real-world problems.
4. The writing of the paper should be significantly improved. There are even missing references, such as "... This sparsifier can be further reduced to O(n log n/ε2) by the existing works []." on page 7.
5. The experimental results are not encouraging: spectral sparsification only produces marginal improvement for a few heterophilic graph datasets but degraded performance for well-known datasets.

**Questions:**

1. What's the percentage of edges that were retained after using the proposed spectral sparsification technique in GNN training?
2. Is there any reduction in the overall GNN training time?
2. How to determine the spectral similarity (\epsilon) in the spectral sparsification step?
3. What is the connection between spectral similarity for spectral sparsification and the final GNN performance (accuracy)?

---

> ### Author Response · Authors · 2023-11-18
>
> Thanks for your time and efforts spent on our work and detailed feedback!
>
> **Reply to weaknesses**
>
> 1. We acknowledge that our explanation of the model's fitting ability on various datasets, particularly heterophilous ones, could be more comprehensive.
> Our approach, which approximates the equivalent propagation matrix of established spectral methods, generally inherits the fitting capabilities of these base models.
> Models like GPRGNN, APPNP, and others, including unimplemented single-layer methods in our work like BernNet, have shown effectiveness on heterophilous datasets.
> Due to space limitations in the main paper, our focus was on performance and scalability.
> However, a more detailed analysis concerning heteophilous datasets and complex filters will be included in the appendix soon.
>
> 2. Our method expands upon the foundational theory established by Cheng et al., extending it to include dynamic (learnable) polynomial coefficients.
> This theoretical backing has enabled us to devise two graph sparsification methods for both static and learnable coefficients, thereby broadening the applicability of spectral methods.
> Additionally, we introduce a node-centric approach to improve efficiency in sparse-labeled graphs, which can be substantiated as a form of spectral sparsification independent of Cheng et al.'s work.
>
> 3. Regarding the hyperparameter setup, our framework introduces only one additional parameter, ```--ec```, compared to the baseline spectral methods.
> This controls the sampling density.
> What may introduce the misunderstanding may stem from our code's simplification from a larger training framework supporting various baselines.
> As clarified in Appendix A.4, only a fraction of the hyperparameters in the file ```train.py``` are employed for training a single model.
>
> 4. We will instantly check and refine any confusion, misinterpretation, or errors in our paper.
> We are committed to integrating all writing suggestions and will conduct a thorough review to enhance the quality and clarity of our paper.
>
> 5. Our model is approximate, and its performance has a connection to the base method used.
> Most current scalable methods, including Cluster-GCN, GraphSAGE, GraphSAINT, LADIES, GNNAutoScale, etc., exhibit some performance loss compared to their base model, particularly as the scale of the benchmark grows.
> These methods typically balance efficiency, performance and applicability.
> Our model not only increases the applicability of spectral methods but also demonstrates comparable or superior performance even against robust base models like GPR-GNN, effectively addressing the issue of performance loss in scalability.
>
> **Reply to questions.**
>
> 1. Our primary goal in sparsification is to approximate the equivalent propagation matrix of spectral methods, denoted as $L_K=\sum_{k=0}^K w_k L^k$, which is inherently a dense matrix.
> Taking the dataset ```twitch-gamer``` as an example, with $K=2$, the non-zeros in $L_K$ amount to $6,827,688,588$.
> In contrast, our approximated graph contains only $12,136,808(1.7\%)$ in GPR-LS and $6,068,404(0.9\%)$ in APPNP-LS with ```--ec=3```.
> The number of sampled edges in our method increases linearly with $K$, while the non-zeros in $L_K$ grow exponentially, with an upper bound of $n^2$. The proportion of retained edges remains below $2\%$ in all cases.
>
> 2. Yes, our method reduces the training time of the base models.
> We will provide a detailed wall-clock time comparison of GNN training in the updated version of our paper, and here is the preliminary results for preview.
>
> | time per epoch (ms) | Cora | Computers | Twitch-gamer | Penn94 |
> | ------------------- | ---- | --------- | ------------ | ------ |
> | APPNP               | 6.50 | 7.59      | 19.99        | 13.75  |
> | APPNP-LS            | 5.58 | 5.83      | 16.73        | 12.42  |
>
> | GPU memory consumption (GB) | Cora  | Computers | Twitch-gamer | Penn94 |
> | --------------------------- | ----- | --------- | ------------ | ------ |
> | APPNP                       | 0.034 | 0.144     | 0.976        | 1.725  |
> | APPNP-LS                    | 0.034 | 0.127     | 0.845        | 1.821  |
>
> 3. Our theoretical framework outlines the asymptotic complexity of the required sample edges.
> In our implementation, we set the number of sampled edges to $n\cdot\log n\cdot ec$, where $ec$ is the hyperparameter controlling the constant of sampling numbers.
> The larger $ec$ is, the more accurate our approximation will be (i.e. smaller $\varepsilon$).

---

> > ### Author Response · Authors · 2023-11-18
> >
> > 4. Our method employs an approximated equivalent propagation matrix for GNN training, enhancing scalability.
> > Since we elevate the scalability, the performance is influenced by the imprecision introduced by the approximation.
> > Generally, the closer the approximation is to the original, the more similar the performance will be to the base models.
> > However, reducing the ```--ec``` value introduces greater randomness in the propagation matrix, potentially affecting performance significantly and influencing the optimal training epoch.
> > GNN training performance remains an empirical aspect of current research.
> >
> > We are grateful for the opportunity to discuss our work further and remain open to addressing any additional queries or points of confusion.

---

### Official Review · Reviewer_aEiE · 2023-11-04

**Soundness:** 2 fair
**Presentation:** 2 fair
**Contribution:** 2 fair
**Rating:** 5
**Confidence:** 4

**Summary:**

The paper proposes a new approach for scaling spectral graph neural networks. Unlike previous efforts that focused on preprocessing feature propagation steps, the proposal relies on Laplacian sparsification, which aims to obtain a sparse graph that retains the spectral properties of the original graph. Experiments using small-scale and large-scale node classification tasks aim to show the effectiveness of the proposal.

**Strengths:**

- The paper tackles the very relevant issue of the scalability of spectral graph neural networks --- the motivation is clear and strong;
- Results on Ogbn-papers100M demonstrate the scalability of the proposed method.

**Weaknesses:**

- Overall, it is unclear if the proposed idea only applies to linear-in-the-parameters spectral GNNs. The definition of spectral GNNs (in the introduction) says they take the form $Y=g_w(L, f_{\theta}(X))$ where $g$ is a polynomial graph filter and $f$ is a linear layer. However, this formulation seems very restrictive and, for instance, does not encompass a simple 2-layer GCN.
- Although the motivation focuses on scalability, the experiments only measure predictive performance. I expected to see an extensive comparison of memory usage and wall-clock time for different methods and datasets. In addition, the paper should report error bars for assessing statistical significance.
- The paper only applies the proposed idea to APPNP and GPR GNNs. I would like to see results for other spectral GNNs (e.g., JacobiConv).
- The theory does not seem particularly useful since implementing GNNs involves non-linearities, rendering gradient estimates biased. Furthermore, results stem almost directly from previous works.

**Questions:**

1. We could also design spectral GNNs by stacking layers of polynomial spectral filters interleaved with ReLU activation functions. Does the proposed approach apply to such models? Would it affect the theoretical analysis?
2. What is the improvement in efficiency by applying the node-wise sampling method (section 3.3)? It would be useful to include some numbers in the Appendix.
3. Is the sampling (sparsification) procedure applied at each forward pass or only once before training?
4. The statement of Theorem 4.1 seems to be an imprecise version of Theorem 2.2 of Cheng et al. 2015 --- it is unclear what is the random variable in the modified statement.


Minor comments/suggestions:
1. The sentence "while keeping the number of non-zeros within an acceptable range" in Contribution is unclear. I would briefly explain the idea behind Laplacian sparsification in the introduction for clarity. One or two sentences should be enough.
2. I think the claim '[...] which is the first work tackling the scalability issue of spectral GNNs' is misleading since GCN can be viewed as a spectral GNN, and other works (e.g., SGC, LanczosNet) have tackled scalability issues of GCNs.
3. There is a significant overlap of ideas in section 3.1 and 1. I suggest reducing the overlap for readability.
4. Please point out the exact Theorem in the paper (Cheng et al. 2015) when saying: 'We have extended the original theorem proposed by (Cheng et. al, 2015) ...'. Also, I suggest creating a specific subsection to prove Theorem 3.2 (as you have done for Theorem 4.3) --- I found the discussion in A.1 overloaded.
5. Some notation is introduced in Algorithm 1, such as e_u and e_v. Is  e=(e_u, e_v) in step 1 of Algorithm 1?
6. 'Some of the early classic GNNs, like GCNs, employ static Laplacian polynomials as the graph filter.' This is questionable since the coefficients of the linear layer can be viewed as multi-head spectral coefficients --- in fact, GCNs were introduced this way.
7. What is $\alpha$ in Section 3.2.1?
8. $w$ has been used to denote both polynomial filter coefficients and weights in weighted graphs (Definition 3.1).
9. The first identity of Eq. (2) should be $L^k$ instead of $L^K$.
10. I would include the main algorithm (node-wise procedure) in the main text (btw, there is no appendix 8).
11. Could you elaborate on the last identity in Eq. (2)? Or provide pointers?

---

> ### Author Response · Authors · 2023-11-18
>
> Thanks for your time and efforts spent on our work and detailed feedback!
>
> **Reply to weaknesses.**
>
> - We will refine any ambiguity previously introduced regarding the application of our concept.
> Our definition of polynomial filters aligns with the broader theme of **representation learning**
> This is evident as the output delineates the learned representation of nodes, adaptable for various downstream tasks.
> For example, in our implementation, we use $Z=f_{\theta'}(\sigma(Y))$ to denote the predicted labels, where $\sigma(\cdot)$ represents the nonlinear activation function.
> The structure of $Y$ aligns with prevalent spectral methods employing a single layer of polynomial filters, such as GPR-GNN, BernNet, JacobiConv, and others.
> We posit that the standard 2-layer GCN diverges from the typical framework of spectral GNNs.
>
> - Our primary experimental findings focus on comparing the performance of our methods against established baselines.
> We observe scalability issues, particularly regarding 1) the substantial memory requirements for graph propagation of raw features in the Penn94 dataset, and 2) the extensive storage demands of the Ogbn-papers100M dataset.
> Our methods demonstrate effective scalability in these contexts, underscoring their advantages.
> A detailed analysis of time and memory consumption will be provided in our forthcoming paper, along with a demo version for preliminary feedback.
> Additionally, we will soon update the error bars in our paper.
>
> - We choose APPNP and GPRGNN for their soundness in the realm of spectral GNNs with static and learnable coefficients.
> Further experiments comparing our methods with other spectral GNNs like JacobiConv are underway and will soon be updated in our paper.
>
> - Our approach is centered on developing an approximation of the equivalent propagation matrix in spectral GNNs, specifically $L_K = \sum_{k=0}^K w_k L^k$.
> While approximation inherently involves some level of randomness and potential inaccuracy, we ensure practical performance by maintaining spectral properties as detailed in Theorem 3.2
>
> Previous studies often rely on subsampling, historical embedding, or some other techniques to achieve satisfactory practical performance.
> However, these methods frequently fall short of providing robust support for gradient estimation.
> The incorporation of linear layers and non-linear activations in these models makes their final performance challenging to analyze, often rendering the results more empirical.
> Our model demonstrates impressive practical performance, thereby validating the effectiveness of our proposed methods.
>
> A novel aspect of our approach is the extension of the original theorem in a more generalized manner, incorporating dynamic (learnable) polynomial coefficients.
> This theoretical foundation has enabled us to devise two versatile graph sparsification techniques, tailored for both static and learnable coefficients, enhancing the applicability of spectral methods.
> Furthermore, we introduce an intuitive node-centric approach to improve efficiency in sparsely-labeled graphs.
>
> **Reply to questions.**
>
> 1. Our approach innovatively replaces the conventional propagation layer with a one-time propagation mechanism, which is also applicable to multiple filter layers.
> To substantiate this, we will present experimental results from several 2-layer spectral models in our paper, and we will present a demo version of our results ahead.
>
> |          | Cora           | Cite.          | PubM.          | Actor          | Wisc.          | Corn.          | Texas          | Photo          | Comp.          | Twit.          |
> | -------- | -------------- | -------------- | -------------- | -------------- | -------------- | -------------- | -------------- | -------------- | -------------- | -------------- |
> | GPR      | 88.80$\pm$1.17 | 81.57$\pm$0.82 | 90.98$\pm$0.25 | 40.55$\pm$0.96 | 91.88$\pm$2.00 | 89.84$\pm$1.80 | 92.78$\pm$2.30 | 95.10$\pm$0.26 | 89.69$\pm$0.41 | 73.90$\pm$0.65 |
> | GPR2L    | 88.09$\pm$1.07 | 80.23$\pm$1.05 | 91.31$\pm$0.39 | 42.08$\pm$1.11 | 92.38$\pm$1.75 | 85.90$\pm$2.62 | 87.87$\pm$2.79 | 94.24$\pm$0.39 | 90.24$\pm$0.35 | 73.71$\pm$0.99 |
> | GPR2L-LS | 88.82$\pm$0.92 | 80.60$\pm$0.63 | 90.98$\pm$0.25 | 42.82$\pm$0.99 | 96.75$\pm$1.25 | 89.51$\pm$2.46 | 91.80$\pm$2.30 | 95.1$\pm$0.42  | 90.57$\pm$0.33 | 73.88$\pm$0.58 |
> | $\Delta$ | **+0.73**      | **+0.37**      | -0.33          | **+0.74**      | **+4.37**      | **+4.61**      | **+3.93**      | **+0.86**      | **+0.33**      | **+0.17**      |
>
> Our theoretical analysis still confirms its effectiveness, independent of the number of filter layers used.
> In fact, those typical spectral GNNs tend to avoid stacking multiple filter layers due to concerns about time and memory overhead, with unequal performance elevation.

---

> > ### Author Response · Authors · 2023-11-18
> >
> > 2. The time of our node-wise sampling methods decreases in datasets with sparse labels.
> > By directly sampling edges from involved labeled sets, our method avoids the wasted sampled edges of the whole-graph training.
> > For example, in the Ogbn-papers100M dataset, where labeled nodes account for only 1.39\%, edge sampling would result in over 99\% of edges being irrelevant as they connect unlabeled nodes.
> > We appreciate the recommendation on this matter and will address it in the upcoming revision of our paper.
> >
> > 3. In our implementation, the sampling method precedes each forward pass.
> > When using learnable coefficients ($\mathbf{w}$), sampling is mandatory for each iteration.
> > However, for static coefficients, this step is optional.
> >
> > 4. We will eliminate any confusion caused by our previous statements.
> > To clarify, $\widetilde{G}$ refers to the sampled graph, while $M$ denotes the upper bound of the number of edges sampled.
> >
> >
> > **Reply to minor comments and suggestions**
> >
> > 1. We appreciate your advice and will revise our statement accordingly, adding necessary explanations for enhanced clarity.
> >
> > 2. We concur with the viewpoint that GCN embodies characteristics of both spectral and spatial GNNs, stemming from its roots in ChebNet and its neighborhood information aggregation mechanism.
> > We acknowledge that popular spectral filters typically operate in the spectral domain with various polynomial bases, represented as $Y=g_\mathbf{w}(L,f_\theta(X))$.
> > While scalability issues in GCN have been extensively explored in models like GraphSAGE, FastGCN, Cluster-GCN, GraphSAINT, and GNNAutoScale, our paper pioneers in addressing the scalability challenges specifically in spectral GNNs with polynomial filters.
> > We will refine our statement for greater accuracy and clarity.
> >
> > 3. Thanks for the suggestion; we will endeavor to minimize our overlap in content.
> >
> > 4. We will pinpoint the relevant theorem and provide a more detailed elaboration on its proof.
> >
> > 5. Yes, and we will ensure this explanation is included in the paper.
> >
> > 6. The term 'static' implies that the coefficients $\mathbf{w}$ in $g_\mathbf{w}(\cdot)$ are not learnable and remain constant during training.
> > The linear layer is, however, actively trained.
> >
> > 7. The variable $\alpha$ represents the probability of halting at the current node during a random walk.
> > This will be clarified in our paper.
> >
> > 8. We will rectify any symbol misuse immediately.
> >
> > 9. Thanks for the corrigendum. We will fix this instantly.
> >
> > 10. All algorithms discussed in sections 3.2.1, 3.2.2 and 3.3 are our original contributions.
> > Due to space constraints, we have provided limited pseudo-codes, focusing on 'Static Laplacian Sparsified Graph Construction'.
> > We will correct the erroneous reference to the Appendix.
> >
> > 11. This comes from a natural substitution of symbol $\mathbf{P}$, leading to the following derivation:
> >
> > $$
> > \sum_{k=0}^K w_k \mathbf{P}^K =
> > \sum_{k=0}^K w_k \left(\mathbf{D}^{-1/2}\mathbf{AD}^{-1/2}\right)^k =
> > \sum_{k=0}^K w_k \mathbf{D}^{1/2}\left(\mathbf{D}^{-1}\mathbf{A}\right)^k \mathbf{D}^{-1/2}
> > $$
> > $$
> > =\mathbf{D}^{-1/2}\cdot\mathbf{D}\left(\sum_{k=0}^K w_k\left(\mathbf{D}^{-1}\mathbf{A}\right)^k \right)\mathbf{D}^{-1/2}.
> > $$
> >
> > We are grateful for your detailed and constructive feedback and look forward to further discussions on any aspects that may still be unclear or of particular interest.

---

> > > ### Comment · Reviewer_aEiE · 2023-11-21
> > >
> > > I have read the responses to all reviewers, and I appreciate the effort put into addressing each concern. However, overall, my concerns were not sufficiently addressed, especially the ones regarding the i) lack of assessment of computational performance, ii) validity of theoretical results under multilayer models, iii) novelty of theoretical analysis, and iv) comparison/applicability against/to more baselines.
> > >
> > > I thank the authors for answering my questions and for fixing the minor issues. I hope to see these reflected in the revised version of the paper. Although I am inclined to keep my initial score, I will deliberate after seeing the revised paper (which is not currently available) and discussing it with AC and other reviewers.

---

### Author Response · Authors · 2023-11-18
**Reply to all reviewers**

We extend our deepest gratitude to all reviewers for your insightful and expert evaluations, which have significantly contributed to enhancing the quality of our research.
Addressing your questions and comments is our utmost priority.

Currently, we are conducting several additional experiments to substantiate our findings further.
We assure you that updates to our paper, incorporating these new results and revisions, will be made available in real time.

Your dedicated feedback is invaluable in refining our work, and we look forward to presenting an improved version of our paper that reflects your constructive insights.

---

### Author Response · Authors · 2023-11-20
**Seeking Feedback and Further Guidance**

Dear Reviewers,

As we approach the end of the discussion phase for Submission7522, we note that we have yet to receive any responses from your end. We fully appreciate that thorough and thoughtful reviewing requires time and effort, and we respect this aspect of the process.

However, we are eager to understand if there are any errors, concerns, or areas for improvement in our submission that we can address. Your specialized insights and feedback are invaluable to the development of our work. If you could take the time to review our rebuttal and provide any additional comments, it would be greatly appreciated.

We assure you that all feedback received will be carefully considered and incorporated into the final version of our paper. We are committed to thoroughly reviewing and refining our work based on your guidance.

Thank you very much for your time and dedication to this process. We look forward to your valuable feedback and continuing our dialogue.

Best regards,

Authors of Submission7522

---

### Meta-Review · Area_Chair_T2s9 · 2023-12-09

**Metareview:**

This paper proposes to sparsify Laplacian to keep essential spectral properties of the graph and enable fast and end-to-end training. While the paper presents an interesting approach to improving the scalability of spectral GNNs, it falls short in several critical areas. These include the depth of theoretical innovation, lack of comprehensive experimental validation and comparison with existing methods, issues with clarity and presentation, and practicality concerns due to multiple hyperparameters. The reviewers unanimously agree the paper does not meet the ICLR bar yet. Overall the paper requires significant refinement and deeper exploration to solidify its contributions and practical applicability.

**Justification For Why Not Higher Score:**

Solidity in theoretical analysis and experiments is questioned by multiple reviewers.

**Justification For Why Not Lower Score:**

N/A

---

### Decision · Program_Chairs · 2024-01-16

Reject